# The fecal resistome of dairy cattle is associated with diet during nursing

Jinxin Liu [1,2], Diana H. Taft[1,2], Maria X. Maldonado-Gomez[1,2], Daisy Johnson[1,2], Michelle L. Treiber [1,3], Danielle G. Lemay [3,4,5], Edward J. DePeters[6] & David A. Mills[1,2,7]

Antimicrobial resistance is a global public health concern, and livestock play a significant role in selecting for resistance and maintaining such reservoirs. Here we study the succession of dairy cattle resistome during early life using metagenomic sequencing, as well as the relationship between resistome, gut microbiota, and diet. In our dataset, the gut of dairy calves serves as a reservoir of 329 antimicrobial resistance genes (ARGs) presumably conferring resistance to 17 classes of antibiotics, and the abundance of ARGs declines gradually during nursing. ARGs appear to co-occur with antibacterial biocide or metal resistance genes. Colostrum is a potential source of ARGs observed in calves at day 2. The dynamic changes in the resistome are likely a result of gut microbiota assembly, which is closely associated with diet transition in dairy calves. Modifications in the resistome may be possible via early-life dietary interventions to reduce overall antimicrobial resistance.

[1] Department of Food Science and Technology, Robert Mondavi Institute for Wine and Food Science, University of California, Davis, One Shields Ave., Davis, CA 95616, USA. [2] Foods for Health Institute, University of California, Davis, California, One Shields Ave., Davis, CA 95616, USA. [3] USDA ARS Western Human Nutrition Research Center, 430 West Health Sciences Dr., Davis, CA 95616, USA. [4] Genome Center, University of California, 451 Health Science Dr., Davis, CA 95616, USA. [5] Department of Nutrition, University of California, Davis, California, Davis, CA 95616, USA. [6] Department of Animal Science, University of California, Davis, California, Davis, CA 95616, USA. [7] Department of Viticulture and Enology, Robert Mondavi Institute for Wine and Food Science, University of California, Davis, California, One Shields Ave., Davis, CA 95616, USA. Correspondence and requests for materials should be addressed to D.A.M. (email: damills@ucdavis.edu)

Antimicrobial resistance (AMR) is a worldwide public health threat[1] with increasing incidence of multidrug resistance found in clinical pathogens[2,3]. The global dissemination of antimicrobial resistance genes (ARGs), including emerging resistances to "last resort" antibiotics such as carbapenem and colistin, prompts the study of novel methods to control the bacterial clades that harbor and propagate ARGs[4,5]. In addition to pathogens, commensal bacteria are also reservoirs of ARGs and influence AMR transmission[6,7]. Compared with human applications, antibiotics are used considerably more for livestock. AMR in livestock limits therapeutic options and creates reservoirs of resistance that can be transmitted to humans via the food chain or environmental effluents. Although such transmission may compromise treatment of human infections[8,9], in-depth investigations employing metagenomic sequencing to study the diversity and abundance of ARGs, and in particular the genes possessed by commensal bacteria in food-producing animals (e.g., cattle), remains limited[10].

The reservoirs of ARGs present in food-producing animals commonly confer resistance to clinically important antibiotics[11]. The prevalence of antibiotic-resistant bacteria, such as *Escherichia coli* and *Salmonella* species, in dairy cattle is typically age-dependent with a higher abundance in earlier stages of life (i.e., pre-weaned calf) albeit this pattern is not necessarily associated with recent use of antimicrobials[12,13]. Data on the impact of other factors such as diet (milk vs. solid food)[14], herd size and farm type[15] on the presence of antibiotic-resistant *E. coli* in calves remains inconclusive. In addition, antibacterial biocides and heavy metals, e.g. copper sulfate, are commonly used as additives in animal feed and may impose a long-term selection pressure for AMR[16]. Studies have suggested the co-selection of biocide and antibiotic resistances in agricultural soil, water[17], and swine manure[18]; however metagenomic validation of co-occurrence between ARGs and biocide/metal resistance genes (BMRGs) in cattle during nursing is currently lacking.

In dairy cattle, the rumen microbial community is necessary to convert dietary plant substrates into accessible nutrients and has been studied from birth to adulthood[19,20]. Dietary transitions during the early life (i.e., nursing) of calves drive changes of the gut microbiota[21], including the progressive acquisition of species capable of digesting complex carbohydrates. In a recent study, Auffret et al.[22] tested the influence of diet on rumen functional genes by comparing the rumen contents from concentrate-fed and forage-fed beef cattle. Interestingly, the authors found that the rumen resistome (the collection of all ARGs) composition is significantly related to diet, and that both the diversity and abundance of ARGs were higher in concentrate-fed animals[22]. These findings clearly suggest that diet-driven dynamic change of the gut microbiome has the potential to modify the gut resistome. While the microbial communities of all compartments in the gastrointestinal tract (GIT) of dairy cattle are important[23], bacteria harboring ARGs in the fecal microbiota, are more likely to contaminate the wider environment[24]. Studies have profiled the fecal microbial community of dairy animals from birth[25] to first lactation[26], but the profile of resistome over time has not been characterized. In particular, the relationship between the changing bacterial populations in cattle feces and antimicrobial resistance remains unknown.

To address these knowledge gaps, this study examines the hypothesis that the early developing gut microbiome of dairy calves serves as an initial point of establishment for bovine-associated ARGs and is driven, in part, by dietary changes during early life. To test this, time-series feces from newborn calves ($n = 22$) were collected to understand the accumulation of the microbiome with a subset of the samples metagenome sequenced to explore the resistome. Maternal colostrum from corresponding mothers was also studied to identify potential migration of both bacterial strains and ARGs to dairy calves. The co-occurrence of BMRGs and ARGs was explored to assess the risk of heavy metals and biocides in selecting for resistance in the absence of antibiotics. A time-course comparison of carbohydrate-active enzymes in the fecal microbiome of dairy calves documented the relationship between diet and the resistome over time. These findings provide a comprehensive view of the dairy cattle resistome in early life and shed light on interventions leveraging dietary modifications to mitigate AMR in food-producing animals.

## Results

**The gut microbiota is rapidly assembled in dairy calves.** In order to explore how the bovine resistome changes during early life it was necessary to first profile the taxonomic changes in calves. To assess these dynamic microbiota variances, fecal samples were collected from 22 dairy calves ($n = 484$) over the first 10 weeks of life (Fig. 1a). All DNA samples were initially subjected to ribosomal marker gene sequencing, and ~9 million quality-filtered 16S rRNA gene sequences were obtained with $18992 \pm 5370$ (mean ± s.d.) reads per sample. The data revealed that the fecal microbial community was assembled rapidly with alpha diversity as measured by both Chao1 richness and Shannon index significantly increasing over time (GEE; $P < 0.001$ for both models) (Fig. 1b). Sequencing data also displayed a temporal change in microbial structure (GUniFrac, PERMANOVA test by adonis2; $P = 0.005$) (Fig. 1c, d). Over 60% of the fecal microbiota was represented by the bacteria classified into the families Ruminococcaceae, Lachnospiraceae, and Bacteroidaceae (Fig. 1d). Interestingly, Enterobacteriaceae accounted for ~25% of the fecal microbiota in the first week of life of dairy calves, during provision of colostrum and milk replacer, but the relative abundance of this family decreased significantly afterward to less than 5% (Kruskal–Wallis; $P < 0.05$) (Fig. 1d). Although dairy calves had constant access to a calf starter diet from day 2, milk replacer served as the primary source of nutrients and energy during the first few weeks of life, but as time passed calves ate an increasing amount of calf starter (Fig. 1a). In summary, the fecal microbiota of dairy calves increased in diversity over time as the animals aged and their diets became more diverse.

Bovine colostrum, the first food given to calves, serves as a reservoir of bacteria[27,28], and likely contributes to seeding the early-life calf gut microbiota. To examine this, colostrum samples collected from dairy cows ($n = 22$) were also sequenced. After quality filtration, a total of 250184 16S rRNA reads were obtained for 44 colostrum samples with $5686 \pm 427$ (mean ± s.d.) sequences per sample. The first and second colostrum that the calves were given during the first day of life, exhibited similar alpha diversities (Shannon index; Kruskal–Wallis; $P > 0.05$) (Supplementary Fig. 1a) and indistinguishable composition based on beta diversity measurements (GUniFrac; PERMANOVA; $P > 0.05$) (Supplementary Fig. 1b). Colostrum was dominated by bacteria classified into the families Streptococcaceae, Enterobacteriaceae, and Enterococcaceae, representing a combined ~90% of the microbiota (Supplementary Fig. 1c). A substantial portion (~30.6%) of the early dairy fecal microbiota (week 1) were comprised of these families (Fig. 1d) which suggests bacterial transmission.

To further test the possibility of bacterial transmission from colostrum to dairy calves, we employed shotgun metagenomic sequencing on paired feces and colostrum samples randomly selected from the larger cohort. Fecal samples from four time points were included: day 2, day 5, week 3, and week 7. Metagenome binning has the tendency to group closely related species and subspecies[29,30], making it a less than ideal approach

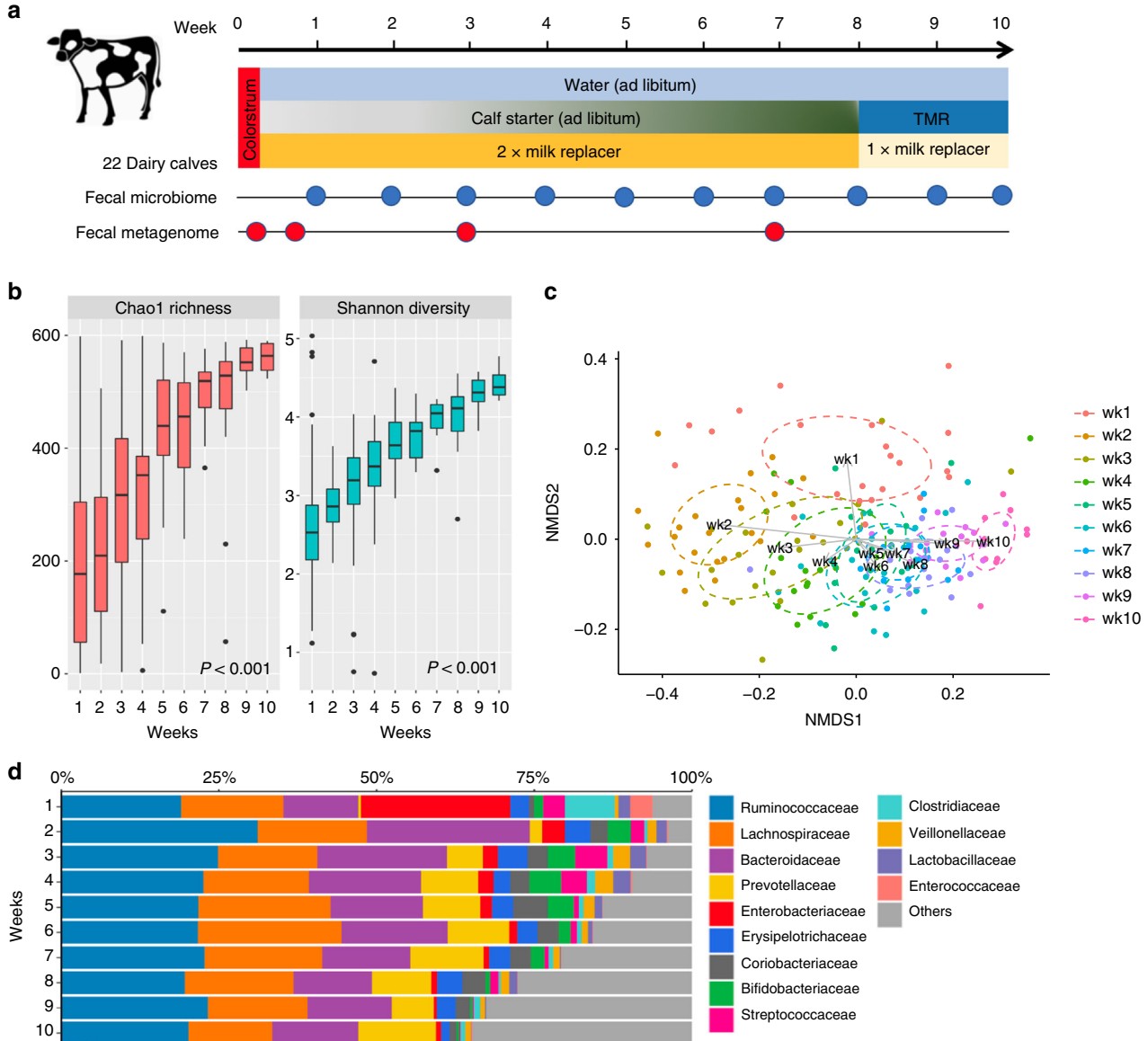

**Fig. 1** Dairy calf fecal microbiota changes over the first 10 weeks of life. **a** The diet, sampling, and sequencing scheme for dairy calves (n = 22). TMR indicates a total mixed ration diet. **b** Boxplots of alpha diversity as measured by Chao1 richness and Shannon diversity index. The Chao1 richness and Shannon index values were presented as the median (central black horizontal line); the lower and upper hinges correspond to the 25th and 75th percentiles. Outliers are displayed as small black circles. **c** 2-dimensional NMDS of dairy fecal samples based on GUniFrac (stress = 0.16). Subsampling to obtain equal sample size (n = 22) at each time point was completed prior to performing the ordination analysis and PERMANOVA. The centroid of each ellipse represents the group mean, and the shape was defined by the covariance within each group. **d** Bar plot depicting the relative abundance of bacterial families over time; unclassified taxa and bacterial families which has a relative abundance less than 1% were grouped into "Others"

to evaluate bacterial transmission at the strain level. Therefore, we performed metagenome strain profiling of the species *E. coli* using two independent approaches, PanPhlAn[31] and StrainEst[32]. *E. coli* was chosen for this analysis because Enterobacteriaceae occurred at high relative abundance in feces in dairy calves during the first days of life and in colostrum; *E. coli* represents the majority of sequences classified in this family (see Methods). The PanPhlAn results indicate that the *E. coli* strains from this study have similar genetic profiles and thus were clustered together amongst all the strains analyzed, including 118 *E. coli* reference genomes (Supplementary Fig. 2a). In particular, the dominant *E. coli* strains from colostrum and fecal samples at day 2 showed similar functional capacities (Jaccard; PERMANOVA; P = 0.15) (Supplementary Fig. 2a). At day 5, in the fecal samples of dairy calves, dominant *E. coli* strains started to diverge from colostrum

samples which eventually demonstrated distinct genetic profiles at week 3 (Jaccard; PERMANOVA; P = 0.004) (Supplementary Fig. 2a).

In addition, StrainEst data suggests that there was a greater diversity of co-existing *E. coli* strains in dairy calves at day 2 compared to colostrum samples, and this diversity increased to day 5 but decreased dramatically at week 3 (Friedman's test; P = 0.02) (Supplementary Fig. 2b). On average, there were 63 distinct *E. coli* strains observed in colostrum samples, with 33.13% of them were shared with dairy calves at day 2. When looking at the overlap between colostrum samples, day 2, and day 5 samples, 15.63% of strains were found in all three samples. By week 3, only one of the three calves included in the metagenomic sequencing subset had high enough levels of *E. coli* detected for the strain analysis. In this calf, 3.08% of *E. coli* strains in colostrum were in

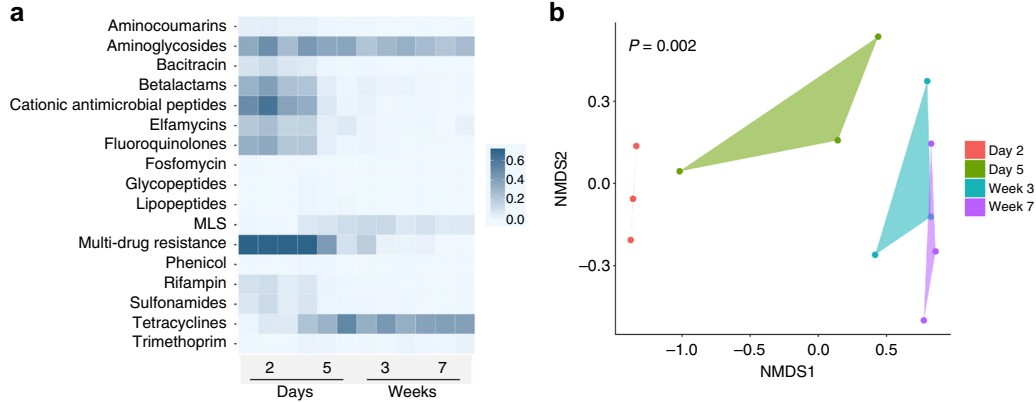

**Fig. 2** The fecal resistome structure of dairy calves over time. (**a**) Dairy calf fecal samples exhibited a broad antibiotic resistance spectrum including resistance to 17 classes of antibiotics. The counts data were normalized by 16S rRNA gene and the ARG abundance was expressed as "copy of ARG per copy of 16S rRNA gene" (see Methods). **b** 2-dimensional NMDS of calf resistome based on a Bray-Curtis dissimilarity calculation (stress = 0.02), showing changes in resistome structure over time assessed by PERMANOVA. Polygons were applied to group samples at the same time point. Source data are provided as a Source Data file

the day 2, day 5, and week 3 fecal samples (Supplementary Fig. 2c). For both strain profiling analyses, *E. coli* was below our detection limit at metagenomes from week 7. This is likely due to the gradually increased gut microbiome diversity and decreased relative abundance of *E. coli* over time. Overall, our findings clearly suggest that colostrum acts as a carrier of specific bacterial species that seed and temporally persist in the gut microbiome of dairy calves.

**Fecal resistome significantly changes over time in dairy calves.** To understand the resistome dynamics during early life, shotgun metagenomic sequences used for strain analysis were tested for the presence of ARGs. Sequencing sample size ($n = 3$ dairy calves) was determined based on the fact that dairy calves of the same breed in our cohort have limited genetic diversity, and they were housed in the same way with identical diets, more importantly, at the same age they possess very similar microbiota (16S rRNA gene sequencing analysis) and thus we expect small resistome variance between subjects. 135 Gb of Illumina sequencing data were produced from 12 fecal microbiomes of dairy calves, and we acquired 11.2 million 150 bp paired-end reads per sample (on average) after host-subtraction and trimming. Our analysis with the AMR++ pipeline (see Methods) revealed that dairy calves are a source of 329 ARGs belonging to 139 ARG groups (Supplementary Fig. 3, Supplementary Data 1), which represent 39 antibiotic resistance mechanisms. The detected ARGs were predicted to confer resistance to a collection of 17 classes of antibiotics in an abundance range of 0–3.75 copies of ARG per 16S rRNA gene among samples (Fig. 2a). Over 50% of the ARGs observed in the day 2 samples were predicted to confer a multidrug resistance phenotype; however, the relative abundance of these ARGs decreased afterward. In contrast, tetracycline resistance gradually increased over time reaching a relative abundance above 70% of all ARGs at week 7 (Fig. 2a). ARGs predicted to confer resistance to aminoglycosides represented a substantial portion of the resistome (~30% of total ARGs), and their abundance was relatively stable throughout the 7 weeks included for analysis (Fig. 2a). In general, we observed that the fecal resistome of dairy calves significantly changes over time (Bray-Curtis; PERMANOVA; $P = 0.002$), with early-life samples (day 2 and day 5) forming separate clusters while later samples (week 3 and week 7) were more similar (Fig. 2b).

In addition to the progressive change in the structure of the fecal resistome, the total abundance of observed ARGs decreased significantly over time; starting at 5.14 copies of ARG per 16S rRNA gene at day 2, and then declined to 0.77 copies of ARG per 16S rRNA gene at week 7 (Friedman's test; $P = 0.02$) (Fig. 3a). As expected, we observed small resistome variances between calves and the sequencing sample size did offer sufficient statistical power to capture the dynamic changes of ARGs abundance over time (effect size is 6.68-fold decrease of ARG abundance from day 2 to week 7). In contrast to the progressively reduced overall abundance of ARGs, the number of ARG types (ARG richness) present in dairy calves increased significantly from day 2 to day 5 and declined gradually afterward (Friedman's test; $P = 0.04$) (Fig. 3b).

By assigning the taxonomy to ARG-containing metagenomic contigs, we were able to predict the bacterial origin of observed ARGs. ARGs detected were predicted to belong to 75 different bacterial families, with families Enterobacteriaceae, Lachnospiraceae, Enterococcaceae, Ruminococcaceae, Bacteroidaceae, Streptococcaceae, and Clostridiaceae representing 69.45% of the total ARG abundance and accounting for 96.66% of all detected ARGs (Fig. 3c). Among these microbial clades, Enterobacteriaceae harbors the most ARGs and the abundance of ARGs assigned to this family gradually decreased over time (Fig. 3c). Over 90% of the microbes classified in Enterobacteriaceae were *E. coli* (Supplementary Fig. 4). Consistent with the overall dynamics of observed ARG richness, the number of different ARGs was highest at day 5 for all bacterial families and showed a gradual reduction afterward (Fig. 3d).

Given the prevalence of ARGs potentially contributed by Enterobacteriaceae, we assessed the absolute abundance and explored the resistance profile specifically for this family. The absolute population of Enterobacteriaceae, as determined by qPCR, remained relatively constant throughout the sampling period (Supplementary Fig. 5a), however the relative abundance as determined by amplicon sequencing (Fig. 1d and Supplementary Fig. 5b) showed a dramatic reduction. This reflects the expected increase in total bacterial population during this early stage of gut microbiome development. Interestingly, the abundance of ARGs within the Enterobacteriaceae decreased after week 1 during our sampling period (7.6-fold decrease from week 1 to week 7; Supplementary Fig. 5c) despite the persistent absolute abundance of this group throughout.

Importantly, ARGs conferring resistance to MLS (macrolides, lincosamides, and streptrogramin A and B) and tetracyclines—two medically-important classes of antibiotics—increased within

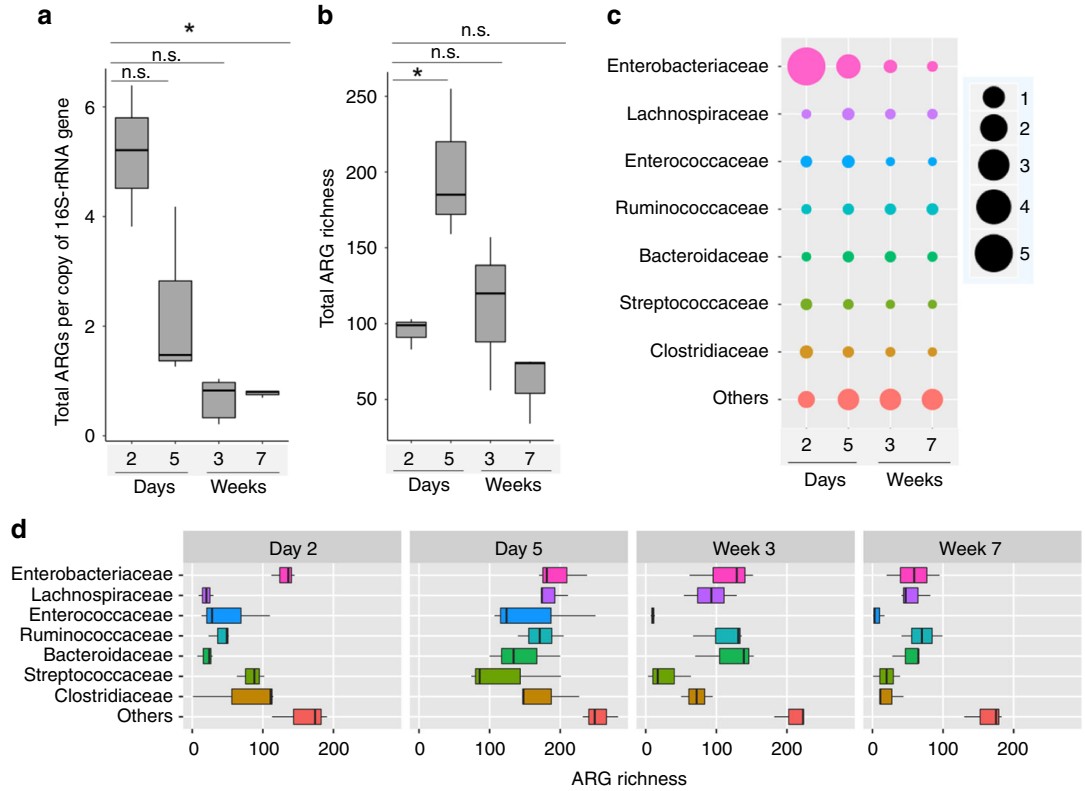

**Fig. 3** The dynamics of the dairy calf fecal resistome over time. **a** Boxplot (boxes representing IQRs with median shown in black) of the distribution of total ARG abundance in dairy calves from day 2 to week 7. **b** Boxplot of the number of unique ARGs observed in dairy calves over time. **c** The predicted bacterial families of ARGs and associated resistance gene abundance in dairy calves. The first 7 families, which are predicted to represent 69.45% of the total ARG abundance and 96.66% of the ARG diversity, are listed here, and the remaining bacterial taxa are grouped into the "Others" category. The taxonomy of ARGs was estimated by assigning taxa at the family level to ARG-containing metagenomic-assembled contigs (see Methods). The size scales with the normalized abundance of ARGs (per copy of 16S rRNA gene) assigned to the particular family at each time point, and colors were used to distinguish different taxa. **d** The predicted ARG richness of each bacterial family over time. The richness is defined as the number of unique ARGs detected during the AMR++ analysis (see Methods). *$P < 0.05$, and n.s. indicates $P > 0.05$ by Friedman's test followed by multiple pairwise comparisons using Nemenyi post-hoc test

the first week but remained high at day 5 until the end of sampling (Friedman's test; $P < 0.05$) (Fig. 4). In particular, tetracycline resistance genes *tet32*, *tet40*, *tetO*, *tetQ*, and *tetW*, and MLS resistance genes including *ermB*, *lnuC*, and *mefA* showed a gradual increase in abundance from day 2 to week 7 (Supplementary Fig. 6). As a separate validation to metagenomic sequencing, quantitative PCR confirmed that the absolute abundance of *tetQ*, one highly abundant tetracycline resistance gene, increased over time. Specifically, a 900-fold increase was observed from day 2 to week 7 in this cohort (Supplementary Fig. 7). This qPCR approach confirmed our resistome findings obtained from metagenomic sequencing across a larger number of samples ($n = 41$) from additional calves.

Transferrable ARGs, which greatly contribute to the overall dissemination of antibiotic resistance, are of specific concern. To assess the prevalence of these genes in dairy calves, assembled metagenomic contigs were run through ResFinder, a database which focuses on acquired ARGs (see Methods). A total of 67 ARGs with transfer potential were detected in dairy calves, conferring resistance to 10 classes of antibiotics (Fig. 5). The observed ARGs were predicted to exhibit a wide distribution across 23 families of bacteria; Enterobacteriaceae, Enterococcaceae, Peptostreptococcaceae, and Streptococcaceae were the families have the most ARGs (Fig. 5). Interestingly, a recently-discovered gene, *optrA*, which confers transferable resistance to oxazolidinones and phenicols[33], was detected in one of the dairy calves at day 5. This gene was predicted to originate from a *Staphylococcus sciuri* and has not been reported from an animal

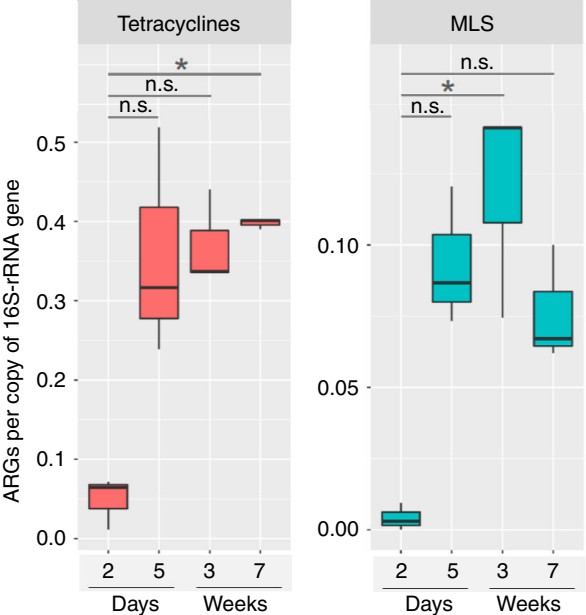

**Fig. 4** The distribution of resistance to the class of MLS and tetracyclines. The analysis was based on the normalized ARGs data from the AMR++ pipeline (see Methods). *$P < 0.05$, and n.s. indicates $P > 0.05$ by Friedman's test followed by multiple pairwise comparisons using Nemenyi post-hoc test. Boxplots representing IQRs with median shown in black

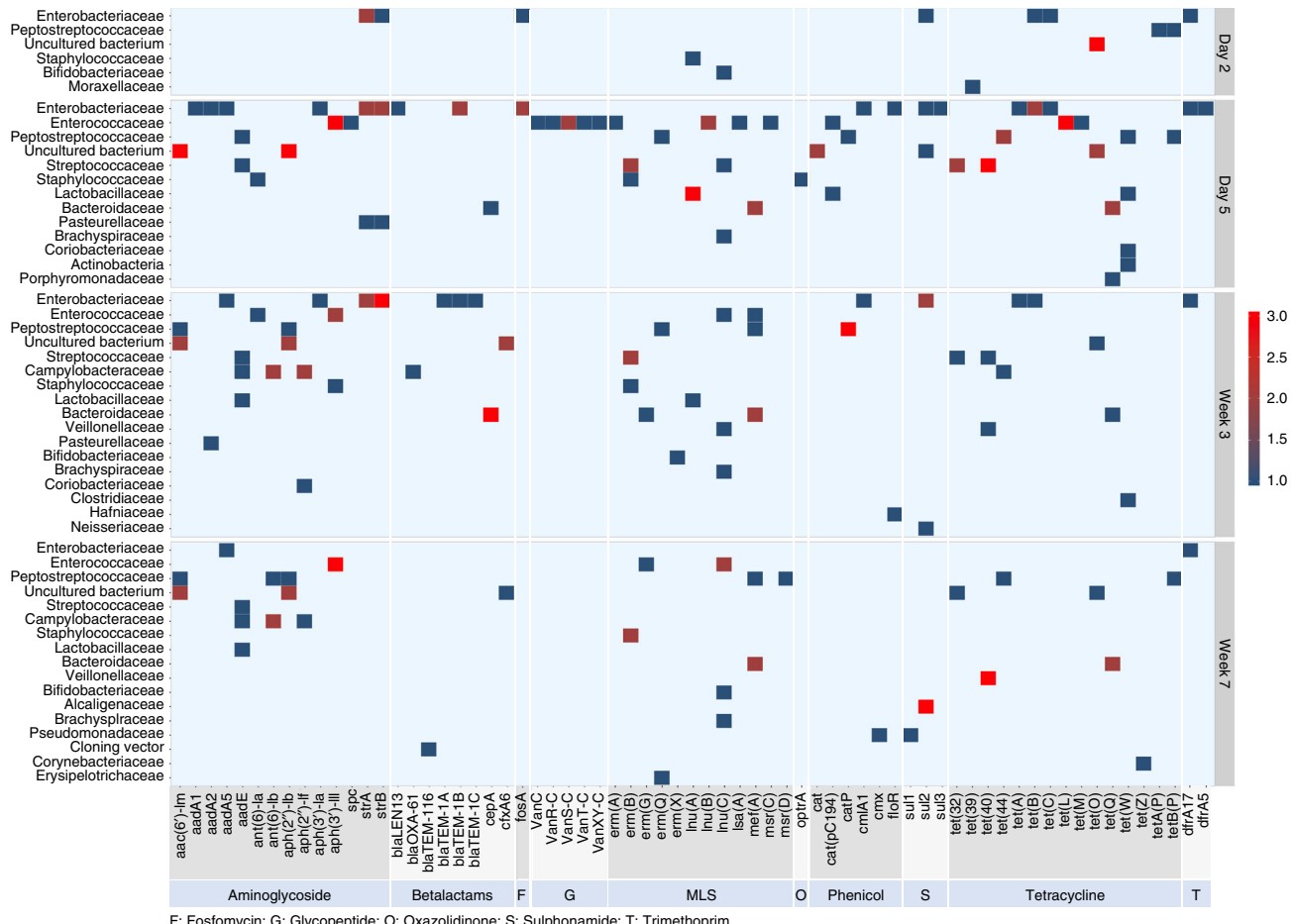

**Fig. 5** The predicted bacterial families of 67 transferrable ARGs. ARGs were grouped per class of antibiotics. Bacterial families were ranked, top to bottom, by diversity of ARGs present, with the families harbor the most diversity of ARGs at the top. Data were obtained from the ResFinder analysis

source in the United States (Fig. 5). In agreement with the total ARGs prevalence analysis with the AMR++ pipeline, an overall increased diversity of transferrable ARGs were observed in dairy calves over time, with day 5 samples enriched with the highest number of different ARGs (Supplementary Fig. 8a). In general, the predicted transferrable ARGs were mostly harbored by two bacterial phyla, Firmicutes (34 genes) and Proteobacteria (28 genes), while the bacterial phyla Bacteroidetes, Actinobacteria, and Spirochaetes possessed a lesser number of ARGs (4, 5, and 1 genes, respectively; Supplementary Fig. 8b). Only limited ARGs were shared across phyla (Supplementary Fig. 8b), indicating that bacterial phylogeny shapes the gut resistome in dairy cattle. For example, the gradually increased ARG, *tetQ* (Supplementary Figs. 6 and 7), was mainly predicted in Bacteroidaceae (Fig. 5).

**Colostrum seeds >90% of early-life ARGs in dairy calves.** Given that dairy calf fecal and colostrum samples share several ARG-enriched taxa (e.g., *E. coli*) and the abundance of ARGs in feces reached its highest levels at day 2 (very early stage), we hypothesized that colostrum serves a significant initial vector of ARGs to calves. To assess this, shotgun metagenomic sequenced colostrum samples were analyzed for the presence of ARGs (see Methods). We produced 38.6 GB Illumina sequencing data from 6 colostrum microbiomes, and ~1.5 million 150 bp paired-end reads per sample were obtained. As expected, of the 105 ARGs detected in colostrum, 73 were present in feces, representing ~90.1% (73/81) of total ARGs detected in dairy calves (Fig. 6a). The overall resistome structure in early-life feces (day 2) closely

matched the pattern in colostrum (Fig. 6b). Fecal samples indicated an overall higher abundance of observed ARGs than colostrum, but colostrum samples showed a higher number of different ARGs present (Fig. 6a, b). In all instances, there was a strong and significant correlation of the resistome pattern between paired colostrum-feces samples (Spearman's $rho = 0.62$ to 0.75, $P < 2.2e-16$) (Fig. 6c). These data support findings presented earlier (Supplementary Fig. 2) suggesting that colostrum serves as a carrier for ARGs.

**BMRGs co-select with ARGs.** Animals enrolled in this study were not treated with any traditional antibiotics. However, exposure of cattle to other compounds, such as antibacterial biocides and metals, may contribute to the promotion of antibiotic resistance through co-selection[16]. Thus, understanding the biocide/metal resistome structure will help illuminate possible sources of indirect selective pressure for antibiotic resistance in dairy calves. Metagenomic sequencing reads of fecal samples were aligned to the BacMet database, matches were tallied, and normalized data were used for comparisons (see Methods). A total of 104 MRGs (Supplementary Fig. 9, Supplementary Data 2) were detected in dairy calves conferring resistance to a series of 15 different antibacterial metal compounds including Sb, As, Cd, Cu, and Ag (Supplementary Fig. 10). In addition, 34 genes found in samples from dairy calves were predicted to confer a biocide resistance phenotype in the present study (Supplementary Fig. 11, Supplementary Data 2). Similar to the dynamics of ARGs in dairy calves, we observed diminishing BMRG abundance with time

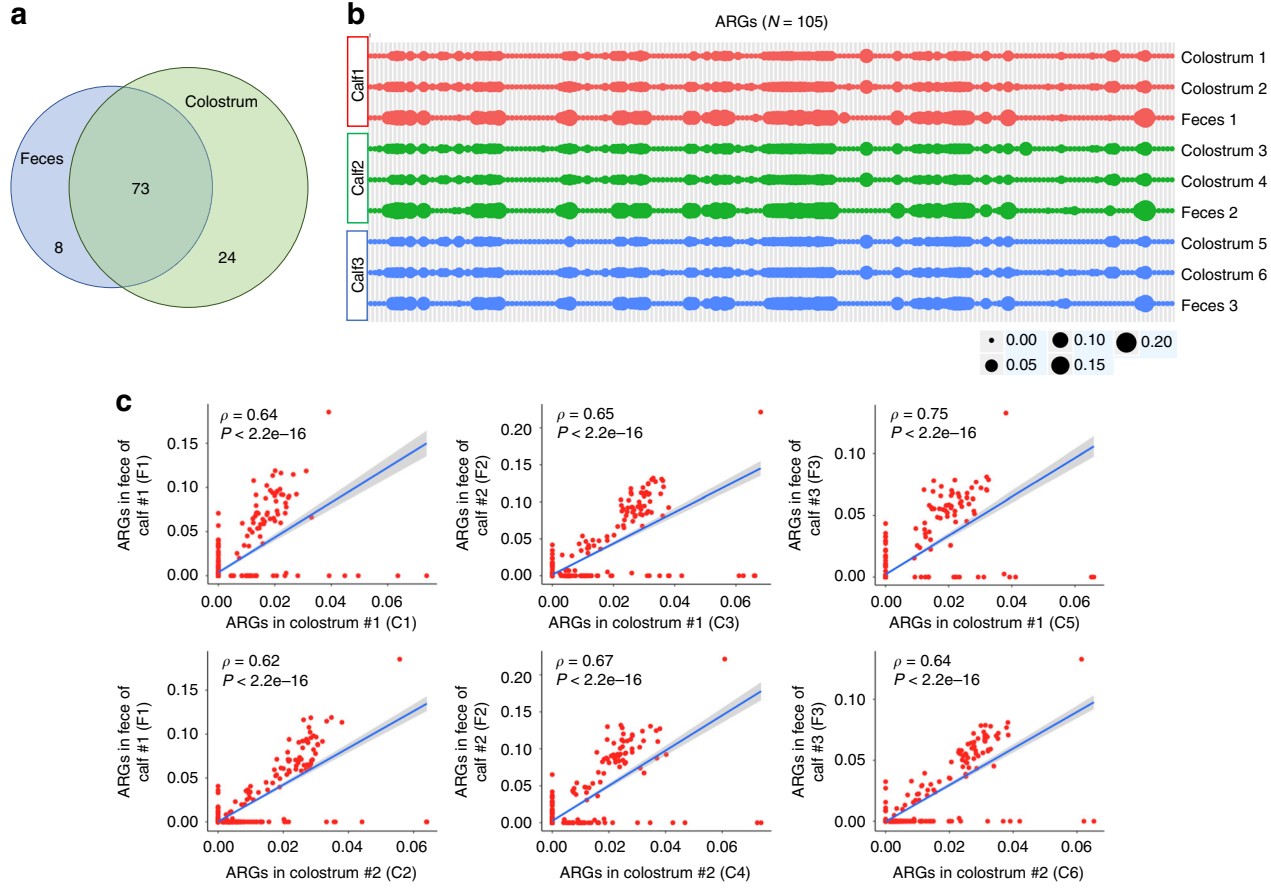

**Fig. 6** The vertical transmission of ARGs from colostrum to dairy calves. **a** Venn diagram showing shared ARGs between colostrum and day 2 fecal samples. **b** The presence of 105 normalized ARG groups (per AMR++ pipeline) was denoted as dots in colostrum and day 2 fecal samples, the size of dots indicates abundance, and the color of lines indicates animals. **c** Spearman's rank correlation coefficient was calculated to study the correlation of ARG-patterns in paired colostrum and fecal samples. C represented colostrum sample, F indicated fecal sample; two colostrum samples and one fecal sample were collected per calf for this analysis, numbers were added to distinguish samples across animals. Source data are provided as a Source Data file

(Friedman's test; $P < 0.05$) (Fig. 7a), but the number of different resistance genes did not change significantly over time (Friedman's test; $P > 0.05$) (Fig. 7b). In summary, the abundance of BMRGs, together with ARGs, decreased over time, in concert with a relative reduction in Enterobacteriaceae.

To assess the potential for co-selection[34], the ARGs and BMRGs with transfer capability that were present in at least half of the metagenomic sequenced fecal samples, were used to predict co-occurrence patterns. Network inference modeling[35] revealed that ARGs and BMRGs were strongly correlated in dairy calves. In the predicted correlation model, 40 resistance genes (nodes) formed three separate clusters while one tetracycline resistance gene, *tet44*, did not show strong correlation with any other genes (Fig. 8). In cluster #1, several biocide resistance genes (*ttgH*, *ttgI* and *ttgG*) co-occurred with the copper resistance genes *cusC*, *cusB* and *cusA*, and all demonstrated a strong connection to a variety of betalactam ARGs (*mecA*, *blaEC*, *ampH*, and *AmpC*). In cluster #2, another set of copper resistance genes (*actP*, *copR*, *pcoR*, and *tcrB*) were positively correlated with Cd and Ag resistance genes. Importantly, the majority of these metal resistance genes exhibited a significant correlation with aminoglycosides ARGs (*aac(6')-Im*, *aph(2")-Ig*, *ant(6)-Ia* and *aph(3')-Ia*) and one MLS ARG (*erm*B). Antibacterial heavy metals are a common "contamination" of animal feed[36], and our analysis hereby indicated that a co-occurrence between ARGs and BMRGs is likely occurring in dairy cattle.

**Changing diet covaries with the fecal resistome in dairy calves.** Because the calf fecal resistome changed significantly during a period of dietary transition, we sought to understand the connection between the modification of the diet-related microbiome and changes in ARG structure. We primarily focused our functional analysis on enzymes involved in the digestion of complex carbohydrates, which represent a large portion of the calves' diet that is inaccessible to the host[37,38]. Fecal metagenomes analyzed for COG and CAZy content revealed a significant change in the functional configuration of fecal microbial communities over time (Bray-Curtis; PERMANOVA; $P < 0.05$ for both cases), with the most substantial differences occurring in the first week (day 2 vs. day 5) (Fig. 9a, b). Specifically, a progressive increase in the number of CAZy enzyme families (CAZy family diversity) was observed (Friedman's test; $P = 0.04$) (Fig. 9c). The dairy fecal microbiome was composed of 6 classes of CAZy-annotated enzymes, of which over 70% characterized CAZy proteins were classified as either a glycoside hydrolase or a glycosyltransferase (Fig. 9d). Of these, the relative abundance of glycoside hydrolases increased after day 2, and that of glycosyltransferase decreased. Genes encoding polysaccharide lyases did not appear in the samples until day 5, and other CAZy-associated enzymes (auxiliary activities) were only present on day 2 and day 5 (Fig. 9d). Differential abundance analysis with DESeq2 revealed that 71 CAZy families significantly changed over time in dairy calves, 50% of which were glycoside hydrolases (Fig. 10a). Furthermore,

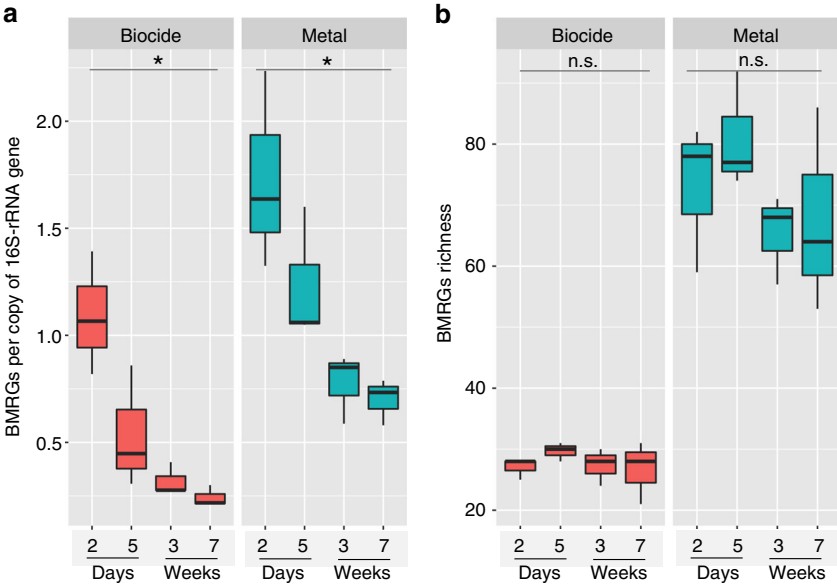

**Fig. 7** The distribution of antibacterial biocide and metal resistance. **a** Boxplot (boxes representing IQRs with median shown in black) of the abundance of normalized BMRGs; BMRGs significantly declined over the course of sampling. **b** Boxplots showing the richness of dairy calves; BMRGs remained comparable over time. BMRGs data were normalized by the same method used for normalization of ARG reads (see Methods). *$P < 0.05$, and n.s. indicates $P > 0.05$ by Friedman's test

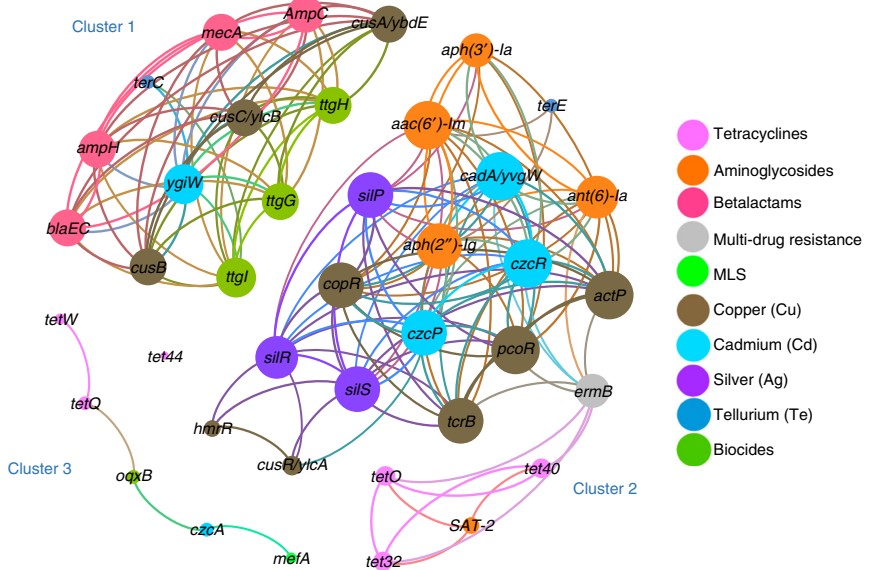

**Fig. 8** The co-occurrence networks among ARGs and BMRGs in dairy calves. The nodes (resistance genes) were colored by corresponding class of antibiotics, metal or biocide compounds, and the size of each node represented the number of connections (degree)

metagenomic assembly was used to retrieve the potential bacterial origin of the enzymes by assigning taxonomy to assembled contigs, and the most abundant taxon at the family level for each significantly changed CAZy family is shown in Fig. 10b. Enterobacteriaceae was predicted to contribute most of the CAZy enzymes that exhibited higher abundance during early days and decreased at later time points (CE8-GH108 and GH127-GT51) (Fig. 10b). Among these enzymes, the family GH1 which includes lactase (EC 3.2.1.108) showed the highest abundance on day 2, reflecting the milk-dominant diet at that time. In contrast, CAZy enzymes, GH20-GH30 and CE6-GH27, which showed higher abundance in later time points were predicted to mostly originate from Bacteroidaceae (Fig. 10b). Two GH families, family GH97, which includes glucoamylase (EC 3.2.1.3) and α-glucosidase (EC 3.2.1.20)), and family GH57, which includes α-amylase (EC 3.2.1.1), increased significantly over time, likely a result of the increased intake of calf starter of which starch serves as the major carbohydrate component (Fig. 1a). The changes in the CAZy enzyme abundances suggest that the decrease in ARGs over time is not merely due to the absence of colostrum, but due to the presence of new carbon sources in the diet that drive abundance of taxa that harbor enzymes which can digest plant polysaccharides and are, coincidentally, also low in overall ARGs.

The predicted taxonomy of significantly changed CAZy enzymes is in agreement with gradual changes observed in the gut microbiota of dairy calves (Fig. 1d), as well as the OTU-OTU

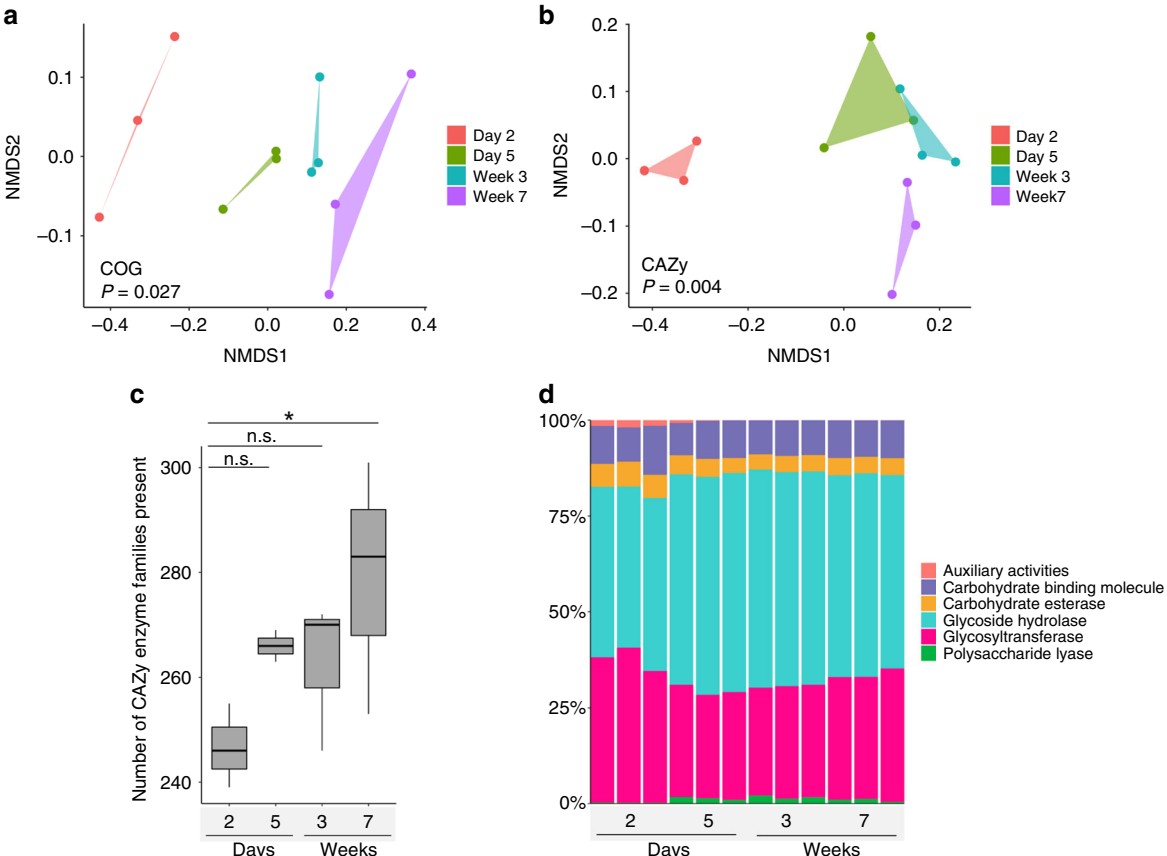

**Fig. 9** Functional capacity of the calf fecal community over time. **a** 2-dimensional NMDS based on Bray-Curtis dissimilarity of COG pathways (stress of 0.07). **b** 2-dimensional NMDS based on Bray-Curtis dissimilarity of CAZy results (stress of 0.1). **c** Boxplots (boxes representing IQRs with median shown in black) of the richness of observed CAZy enzymes (families) in dairy calves from day 2 to week 7. **d** Bar plots of the relative abundance of CAZy families per class of enzymes over time. *$P < 0.05$, and n.s. indicates $P > 0.05$ by Friedman's test followed by multiple pairwise comparisons using Nemenyi post-hoc test. Polygons were applied to group samples of NMDS plot at the same time point

network analysis based on 16S rRNA sequencing data. In this network, the relative abundance of organisms within the families Enterobacteriaceae and Enterococcaceae was negatively correlated with microbes belonging to Bacteroidaceae (Supplementary Fig. 12). Taken together, our data support the hypothesis that diet and time (age) contribute to the taxonomic composition of the gut microbiota in dairy calves, and this influences the fecal resistome.

## Discussion

The early development of gut microbiota is associated with growth in pre-weaned calves, is important for optimal calf performance post-weaning, and is believed to contribute to the profitability of the cattle enterprise[39]. However, young animals typically have both a higher prevalence and a higher abundance of antibiotic-resistant bacteria than adult animals[13,40]; thus understanding the early acquisition and scope of AMR in calves during nursing is of great importance. This work, together with other studies[25,26], documents the dynamic changes in the taxonomic composition of the fecal microbiota of dairy calves, revealing an actively assembling gut microbiome during early life. For example, we clearly demonstrated that facultative anaerobes, such as *Escherichia*, *Streptococcus*, and *Enterococcus* (genera which are commonly associated with broad-spectrum ARGs) represented notable portions of the microbiota in dairy calves during early days, but their relative abundance decreased over time. The early arrival of these bacteria is at least partially linked

with colostrum feeding, and their dynamic changes in relative abundance contribute to the gradually modified resistome during nursing. This study examined the fecal microbiota of dairy calves from birth until slightly after weaning, and the microbial community is likely to continue changing until 1 year of age, when it reaches an adult-like (mature) microbiota[26]. Therefore, a more extended period of sampling may be needed to fully track changes in ARGs as the cow matures.

The fecal microbiome of dairy calves was predicted to harbor resistance to 17 classes of antibiotics according to MEGARes (out of the 22 classes of antibiotics present in this database), a fact which is important because fecal shedding is a critical route of AMR contamination to the environment[24]. In addition, the ARGs observed in this study are closely relevant to human medicine as they include resistance to several medically-important antibiotics, e.g. beta-lactams. Specifically, two major classes of plasmid-mediated extended-spectrum beta-lactamases (ESBL), class A betalactamases (e.g. CTX and cepA) and class D betalactamases (e.g. OXA) were detected in this cohort. The overall ARG abundance (0.77–5.14 copies ARGs per 16S rRNA gene), is of the same magnitude as that observed in recently published metagenomes from livestock-associated samples (0.54–3.1 copies ARGs per 16S rRNA gene) which are significantly higher than the prevalence of ARGs detected in sediments, soil and river water[35,41]. The coherence between these results indicates that the present cohort is a typical livestock group, and our findings herein could be potentially generalized to other populations. Consistent with the hypothesis that a high prevalence of AMR is

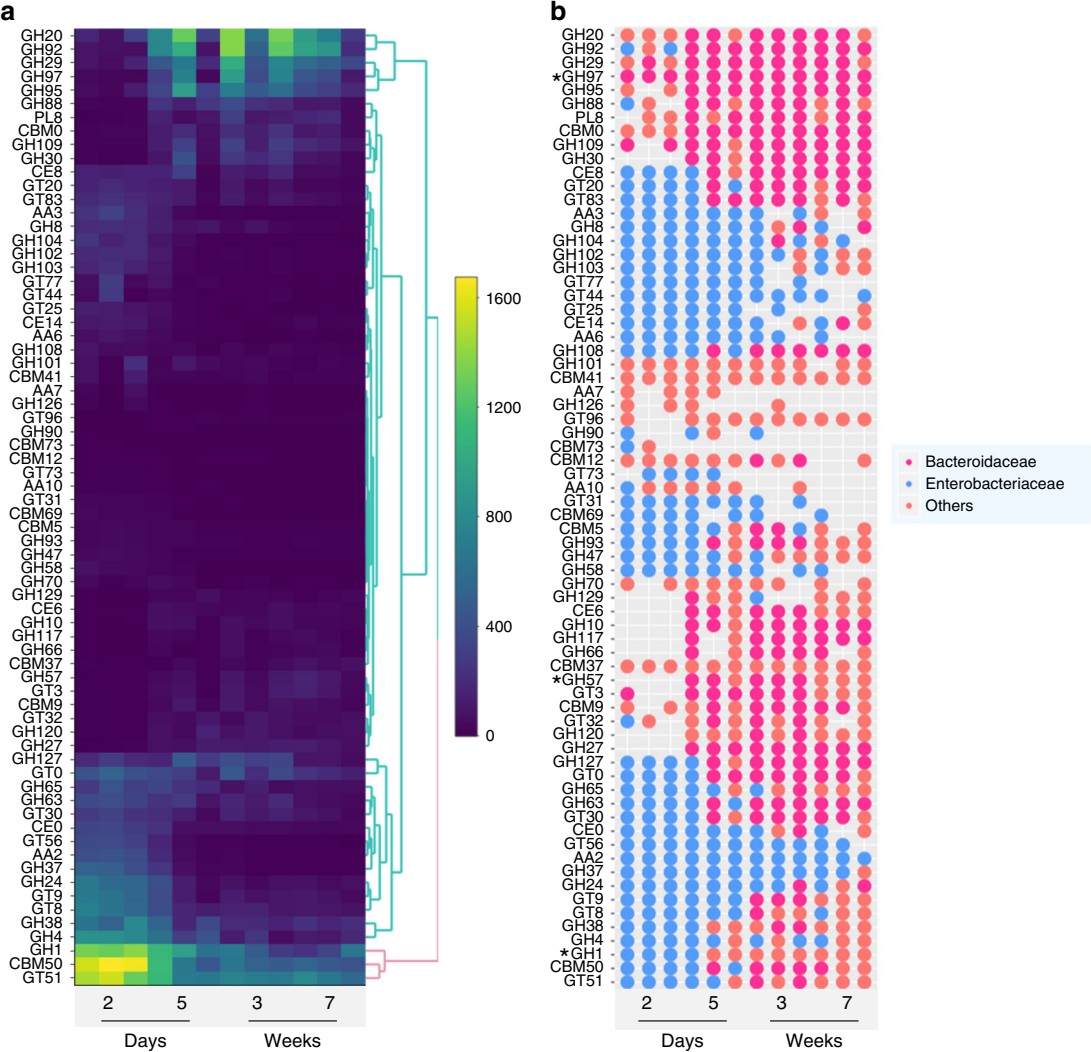

**Fig. 10** The abundance, distribution and predicted bacterial taxonomy of fecal CAZy families. **a** Heatmap depicting the normalized distribution of CAZy enzymes over time, including only differential abundant enzymes ($n = 70$). **b** The most abundant bacterial family predicted to produce the corresponding CAZy enzymes. Dots indicate that the predicted family was either Enterobacteriaceae (blue), Bacteroidaceae (pink), or Others (orange). The absence of a dot indicates that either our inability to predict the bacterial origin or the CAZy family was not detected in a particular calf at a particular time point. Enzymes families GH1 which includes lactase (EC 3.2.1.108), GH97 which includes glucoamylase (EC 3.2.1.3) and α-glucosidase (EC 3.2.1.20)), and family GH57, which includes α-amylase (EC 3.2.1.1) are highlighted with asterisks. Abbreviations: GH, glycoside hydrolases; GT, glycosyltransferases; PL, polysaccharide lyases; CE, carbohydrate esterases; AA, auxiliary activities; CBM, carbohydrate-binding modules. Families CBM0, CE0, and GT0 refer to unclassified CBMs, CEs and GTs, respectively, in CAZy database download 07152016. Differential abundance analysis with DESeq2 revealed that 71 CAZy families significantly changed over time in dairy calves, however, we failed to retrieve the bacterial origin of one CAZy family, GT80, and thus there are a total of 70 CAZy families shown here. Source data are provided as a Source Data file

not necessarily associated with recent antibiotic use[13], animals involved in this study did not receive any antibiotic treatment, but their gut microbiome remained a natural reservoir of ARGs.

Interestingly, dairy calves had an overall reduction (6.68-fold) in the abundance of ARGs as they aged, and this pattern was directly correlated with the decrease in relative abundance of microbial taxa, for example, Enterobacteriaceae, which was predicted to harbor the most ARGs, during early life. The correlated reduction of both ARGs and specific taxa is promising because it suggests that interventions targeting the reduction of these taxa may further reduce the ARG reservoir in the gut of cattle. This possibility is supported by this study and other work, which found the resistome is mainly structured by the bacterial phylogeny, and that Proteobacteria (which includes Enterobacteriaceae) were more likely to carry resistance than other phyla[42,43].

Despite the decrease in total ARGs, we observed an increase in resistance to MLS and tetracyclines antibiotics over time, raising concerns that not every ARG decreases in abundance with age. The increase in MLS and tetracycline ARGs is also consistent with the hypothesis that community membership shapes the resistome because neither MLS nor tetracycline ARGs primarily originate in Enterobacteriaceae. In contrast, tetracycline ARGs, including the highly abundant *tetQ*, were predicted to originate in Bacteroidaceae. While no antibiotics were given to the animals in this study, the increased tetracycline resistance may reflect historical antibiotic use. In the United States tetracycline has been widely used in cattle representing ~30.1% of the total antibiotic usage in food-producing animals[44]. Because MLS is categorized as critically important and tetracyclines as highly important drugs according to FDA[45] the increase of corresponding resistance is a source of concern.

Another approach to assess the risk of AMR is to specifically study the transferrable ARGs (via ResFinder) as these genes are more likely transferred to pathogens and compromise clinical treatment[46]. The results counterbalanced the decreased abundance of total ARGs over time by identifying an increase in the variety of transferable ARGs. Of these ARGs, a recently-discovered multidrug resistance gene, optrA[33], was present in our cohort. This gene confers resistance to several medically-important antibiotics including linezolid, which is often prescribed to treat Gram-positive infections such as vancomycin-resistant enterococci (VRE) and methicillin-resistant Staphylococcus aureus (MRSA). Another gene, fosA, is also of clinical significance because it causes resistance to fosfomycin which is a last-resort antibiotic against carbapenem-resistant bacteria[47]. The increased diversity of transferable ARGs partially reflects the accumulation of new species during community assembly.

Given that enrichment of ARG diversity in dairy calves is undesired, a next logical question is: what is the source of these antibiotic-resistant organisms? Routes of microbial transfer may include vertical transmission from the dam, horizontal transmission from other animals in the herd, or transmission from the housing environment and feed. As a result, colostrum may serve as an important vehicle for bacteria in early life since it is the first feed[27,48]. Doyle et al.[49] indicated that teat surface and herd (i.e., dam) feces are the major contributors of the raw milk microbiota. These data suggest that organisms observed in colostrum are likely environment-specific contaminants. This is supported by the fact that our findings on the bovine colostrum microbiota differed from a recent study in upstate New York which found the samples were dominated by Staphylococcus spp.[28]. Moreover, the dominant taxa in colostrum within our cohort are taxa expected to contain high levels of ARGs, suggesting that colostrum serves as a vector of ARGs transmitted to dairy calves. This inference was confirmed by metagenomic data at two levels. Metagenome strain profiling revealed that the dominant E. coli from colostrum and fecal samples at day 2 have very similar functional capacity (PanPhlAn), and 33.13% of the observed E. coli strains in colostrum were shared with dairy calf at day 2 (StrainEst) which suggests bacterial transmission. In addition, the dairy fecal samples at day 2 shared ~90% ARGs with the colostrum resistome and the distribution of ARGs show very similar patterns in these two samples types. In general, colostrum samples had a greater diversity of ARGs than fecal samples, but shared ARGs showed relatively higher abundance in the feces suggesting that the gut provides better conditions for the proliferation of these ARGs and the microbes that carry them.

Although there was no direct antibiotic use in this cohort, heavy metals and antibacterial biocides, which are commonly detected in animal feed, could contribute to the promotion of AMR[16]. Co-selection of antibiotic and metal/biocide resistance, which may be driven by either cross- or co-resistance, is critical mainly because it can promote and maintain the high prevalence of AMR in the absence of antibiotics[50]. Network inferences from the present study indicated a correlation among ARGs and BMRGs in dairy calves, and resistance genes harbored in the same taxa[42] or co-existing on a transferrable plasmid[51] can lead to a co-occurrence of these genes. While the strain-level proof of the network analysis linking ARGs and BMRGs will need further validation via an independent approach, the metagenomic data certainly suggest that co-selection of antibiotic and metal/biocide resistance is occurring in livestock.

To understand the mechanism responsible for the dynamic change of fecal resistome, and to further understand the relationship between dietary transition and gut microbiota assembly, we examined the functional changes of the gut fecal community in dairy calves. Our findings greatly reflect the connection between diet and functional capacity. Specifically, the CAZyome analysis revealed that the composition of carbohydrate associated enzymes in dairy calves changed significantly over time. The early diet transition from colostrum to milk replacer with a gradual increase of consumption of calf starter mirrors the decrease of lactose associated enzymes (e.g., lactase) and the increase of enzymes related to starch and other complex carbohydrates (e.g., amylase). The CAZy families which were more abundant in early days mostly originated from Enterobacteriaceae while the enzymes that increased over time were predicted to be most likely from Bacteroidaceae and others. Intriguingly, the species within Bacteroidaceae were shown to harbor tetQ and other tetracycline resistance genes, suggesting a possible link between diet-driven changes in microbiome composition and the structure of the resistome. The link between bacterial phylogeny and AMR is suggested by the concomitant decrease in the relative abundance of Enterobacteriaceae, and the reduction in ARGs. This is also evidenced by the gradual increase (demonstrated by both meta-genomic sequencing and qPCR data) of tetQ, occurring concomitantly with the increase in Bacteroidaceae. Furthermore, colostrum containing endogenous microbes, seeded the calves with high level of ARGs in early hours, highlighting the potential role of diet as a direct source of ARGs. Diet may also represent an indirect influence on the resistome by enriching certain taxa, such as Bacteroidaceae, in the gut microbiota.

The global spread of antimicrobial resistance compromises clinical treatments for human infections and actions are needed to reduce the environmental transfer of ARGs and preserve the effectiveness of existing antibiotics. This study suggests that pre-weaned dairy calves serve as a reservoir for ARGs with enriched diversity over nursing. While age and diet are clearly associated with a change in the gut microbiota assembly and an overall drop in ARGs, several resistance genes encoding resistance to clinically important antibiotics (i.e., MLS and tetracyclines) increased. Analyses in this study relied on high-throughput sequencing of DNA, representing both live and dead bacteria. The presence of DNA, especially DNA released from dead microbes, does not ensure antibiotic resistance phenotypes. Future studies involving functional characterization of specific ARG phenotypes are needed. Studies suggest that naturally competent bacteria are able to uptake DNA released from dead bacteria[52], which may contribute to the transmission of ARGs, however, the probability of this occurrence is unknown. In order to reduce these antibiotic (and heavy metal) resistance reservoirs during nursing, novel mitigation strategies are needed. For example, control measures might be incorporated into colostrum handling practices to reduce transmission of ARG-containing microbes to dairy calves. Moreover, tailored prebiotic or probiotic applications might serve to displace ARG-containing bacteria and thus reduce the likelihood of further environmental spread.

## Methods

**Animals, sample collection, and DNA extraction.** A cohort of 22 newborn dairy calves (20 Holstein, 2 Jersey) were raised at the UC Davis Dairy Teaching and Research Facility (Davis, CA). Calves were separated from their dams within a few hours of birth and then housed in a separate hutch. They were given two feedings (3 L/feeding with nursing bottle) of high-quality colostrum within the first 12 h followed by milk replacer (Calva Products, Acampo, CA) and a commercial calf starter (Associated Feed & Supply Co., Turlock, CA) until weaning (Fig. 1a). The milk replacer was given twice a day based on weight, and calf starter was provided ad libitum until weaning. Corn, soybean, barley, wheat, oat, and rice are the major ingredients in calf starter and, starch accounts for a 57–77% of this diet[53]. During weaning (week 8–10), supplementation of calf starter was halted, milk replacer was reduced to a single feeding, and a total mixed ration (TMR) diet was introduced. Calves were offered water ad libitum at all times (Fig. 1a). All animals involved in this study were healthy during our sampling period and received no recorded therapeutic or prophylactic antibiotic treatments.

Fresh feces (n = 484) were obtained from dairy calves, from at birth to week 10 between April and September of 2015, by swabbing the rectum with a sterile cotton

swab. Two colostrum samples (~5 mL each), one per colostrum feed, were collected per calf prior to feeding. Sampling was conducted following relevant ethical regulations for animal research under protocol #18540. Fecal samples were then kept on ice after sampling and then stored at −80 °C until extraction using the ZR Fecal DNA MiniPrep kit (ZYMO, Irvine, CA, USA). Approximately 2 mL vortexed colostrum samples were centrifuged at 10,000×g for 10 min to separate cells and fat from whey. The supernatant and the fat layer were removed, and the pellet were kept frozen (−20 °C) until the Zymo DNA extraction[54].

**16S rRNA gene sequencing and data analysis**. The microbiota was profiled by sequencing the V4 region of the 16S rRNA gene for all DNA samples and negative controls. Specifically, we modified the forward F515 primer[55] to include an eight-nucleotide barcode unique to each sample and a two-nucleotide linker sequence: 5′-NNNNNNNN**GT**GTGCCAGCMGCCGCGGTAA-3′. The reverse R806 primer (5′-GGACTACHVGGGTWTCTAAT-3′) was unmodified. PCR reactions were carried out in triplicate in a 15-μL reaction containing 1 × GoTaq Green Mastermix (Promega, Madison, WI, USA), 1 mM MgCl₂ and 2 pmol of each primer. The PCR amplification conditions included an initial denaturation step of 2 min at 94 °C, followed by 25 cycles of 94 °C for 45 s, 50 °C for 60 s, and 72 °C for 90 s, followed by a final extension step at 72 °C for 10 min. Triplicate reactions were combined and purified using a Qiagen PCR purification column and submitted to the UC Davis Genome Center DNA Technologies Sequencing Core for sequencing on an Illumina MiSeq platform (250-bp paired-end).

The resulting reads were merged using PEAR (version 0.9.8) with a minimum overlap of 120 bp, a maximum merged length of 380 bp, and a minimum merged length of 250 bp[56] and merged reads were demultiplexed using FASTX (version 0.0.14) tools[57]. Primers and barcodes were trimmed from reads using cutadapt (version 1.8.3), and then reads were loaded into QIIME[58]. QIIME1 (version 1.9.1) defaults were used except for as follows: operational taxonomic unit (OTU) picking was completed using the SWARM algorithm[59], the representative sequence set was chosen using the most abundant read in each OTU, and the OTU table was filtered to remove singletons and OTUs that only occurred in a single sample. In addition, samples with low sequencing depth including all the negative controls (< 5000 reads) and OTUs without any reads in any of the remaining samples were excluded for further analysis.

Statistical analysis was performed in R (version 3.4.1)[60]. Generalized estimating equations (GEE) model from the geepack package[61] was implemented to test for associations between the sampling time (predictor variable) and the alpha diversity (outcome variable). Beta diversity was visualized based on generalized UniFrac (GUniFrac) distance matrices[62] using non-metric multidimensional scaling (NMDS) in the vegan package. A 2D plot was used if stress was < 0.2; the number of dimensions was increased until a plot with stress < 0.2 was produced. Ellipses shapes/paths were calculated with a function veganCovEllipse from the vegan package in R. Differences in beta-diversity based on UniFrac distance measures were tested using adonis2 in the vegan package after checking for differences in dispersion using betadisper. SPIEC-EASI (SParse InversE Covariance Estimation for Ecological Association Inference)[63] was used to model OTU-OTU interaction with fecal samples (n = 22) at week 2, and the network was visualized at the family level using Gephi (0.9.2).

**Shotgun metagenomic sequencing, metagenome assembly**. Fecal samples (n = 12) from dairy calves (n = 3) at 4 time-points per calf (day 2, day 5, week 3 and week 7), as well as the corresponding colostrum samples (n = 6; two of each calf), were subjected to metagenomic sequencing. This design allows us to characterize the dynamics of resistome in dairy calves over time, with a specific focus on nursing. Dairy calves at the same age had limited variances in the gut microbiota, and bacterial phylogeny greatly shapes the microbial antibiotic resistome[42], and thus we expect a comparable fecal resistome between subjects. Consequently, a number of three calves would allow us to observe the dynamic changes of ARGs abundance over time if any.

The sequencing library was prepared in the UC Berkeley Functional Genomics Laboratory (FGL). Briefly, each sample was sheared using the 150 bp setting of the Diagenode Bioruptor, then purified and concentrated with the Qiagen Minelute cleanup kit. End repair, a tailing of DNA fragments, and adapter ligation were preformed using the KAPA Hyper Prep library kit. Next, 9 cycles of indexing PCR were performed using the KAPA Hi-Fi Hotstart amplification kit. Cleanup and dual-SPRI size selection were completed using AMPure beads. Libraries were checked for quality on the AATI fragment analyzer.

Shotgun metagenomic sequencing was completed using the Illumina HiSeq 4000 with 150 paired-end reads in the Vincent J. Coates Genomics Sequencing Laboratory at the University of California, Berkeley. Because early life and colostrum samples had high levels of host contamination, BMTagger in bmtools (version 1) was used to remove reads aligning to the bovine genome (version UMD3.1) from all samples. The resulting reads were then trimmed using Trimmomatic (version 0.36)[64] and merged using FLASH (version 1.2.11)[65] prior to downstream analysis. To account for the limitations of the existing database in taxonomy profiling of rumen-associated metagenomes[66], sequencing reads were classified using Kraken2[67] to against a custom database including RefSeq[68] and 4941 metagenome-assembled rumen genomes[69]. The relative abundance of *E. coli*

within Enterobacteriaceae was estimated using Bracken[70]. Metagenome assemblies were generated with trimmed but un-merged reads for each sample using MEGAHIT (version 1.0.6) with default parameters[71].

**Metagenomic strain profiling of *E. coli***. To assess the bacterial transmission from colostrum to dairy calves, the microbial composition of both colostrum and feces were characterized at the strain level using two independent pipelines, PanPhlAn[31] and StrainEst (version 1.2.4)[32]. The sequencing reads from the first and second colostrum samples were analyzed together for the colostrum-calf paired strain transmission analysis. PanPhlAn was used to track the dominant strain across samples by identifying a unique combination of genes in the pangenome of a species, and StrainEst was used to determine the number and identity of co-existing strains of a species (i.e., *E. coli*) in a metagenome. Because Enterobacteriaceae harbors the most ARGs (Fig. 3c) and *E. coli* represents the majority of sequences classified in this family (Supplementary Fig. 4), we elected to track the presence of strains belonging to this species in our cohort. The PanPhlAn results were visualized via a NMDS ordination and pairwise permutation MANOVAs (post-hoc test for multiple comparisons) was performed in R with the RVAideMemire package (version 0.9–69–3)[72], and significant differences between groups were evaluated after FDR adjustment for α = 0.05. Statistical analysis of Shannon diversity estimated with StrainEst was performed with a Friedman's test.

**Resistance gene analysis**. Sequencing reads were aligned to the ARG database MEGARes (version 1.0.1)[73] to characterize the resistome structure following the AMR++ pipeline [https://megares.meglab.org/amrplusplus/latest/html/index.html] with minor modifications. Briefly, merged reads were mapped to the database using BWA with default settings[74], and the SAM formatted alignment file was then analyzed through ResistomeAnalyzer (version 1) by setting the threshold to 80% identity for quantification of ARGs [https://github.com/cdeanj/resistomeanalyzer]. The AMR++ pipeline outputs data into gene, group, mechanism and class levels corresponding to the levels of the annotation in the database hierarchy[73]. The gene-level data (e.g., TEM-77, TEM-107, TEM-73, etc.) were used to calculate the ARG richness; normalized data aggregated from the gene-level output to the group (e.g., TEM, OXA, etc.) and class (e.g. beta-lactams resistance) levels were used for heatmap visualization in this study. MEGARes is a manually curated database that consists of a collection of 3824 ARGs with the reference sequences ranging in size from 211 to 4185 bp[73]. To avoid bias related to variation in ARG size, both ARG sequence length and sequencing depth were included in data normalization prior to statistical comparisons. We normalized the counts data by 16S rRNA gene and the ARG abundance was expressed as "copy of ARG per copy of 16S rRNA gene" as suggested by Li et al.[35,75] using the formula:

$$\text{Abundance} = \sum_{1}^{n} \frac{N(ARG - \text{like sequence}) * L(\text{reads})/L(ARG \text{ reference sequence})}{N(16S \text{sequence}) * L(\text{reads})/L(16S \text{sequence})}$$

(1)

where *n* represents each individual ARG (defined by different sequences), N(ARG-like sequence) is the number of reads annotated as *n* specific ARG mapped to MEGARes database, L(reads) is the length of the sequencing reads matching ARG *n*, L(ARG reference sequence) is the length of ARG *n* in the MEGARes database, N (16S sequence) is the number of reads mapping to 16S rRNA bacterial gene determined by METAXA2 (version 2.1.3)[76], and L(16S sequence) is the average length (1432 bp) of a 16S rRNA gene sequence in the Greengenes database. To visualize dissimilarity between samples, NMDS was performed using a Bray-Curtis dissimilarity calculation.

Bacterial origin of ARGs were predicted by assigning taxonomy to metagenomic-assembled contigs harboring antibiotic resistance genes. Specifically, the ARG-aligned sequencing reads (see AMR++ pipeline) were used to align to assembled contigs (from MEGAHIT) with BWA-MEM[74], and contigs which contain ARG sequences were kept for taxonomic assignment. taxator-tk (version 1.3.3)[77], software designed to perform taxonomic analysis of assembled metagenomes, was applied to predict the bacterial origin of specific contigs. In particular, we used taxator-tk with our custom database which includes both RefSeq[68] and 4,941 metagenome-assembled genomes[69] with parameters -a megan-lca -t 0.3 -e 0.01 to assign taxonomy of ARG-containing contigs at the family level.

In addition to the evaluation of all sequencing data described above, an independent analysis was performed on metagenomic reads obtained from dairy calves that were predicted to fall within Enterobacteriaceae (by alignment to RefSeq genomes[68]), The AMR++ pipeline was implemented separately for these reads in order to study the prevalence of ARGs in this particular family.

Transferrable ARGs are typically of greater concern clinically. To specifically assess these genes, a separate ARG database, ResFinder (version 2.1)[46], focusing on acquired ARGs, was used based on assembled contigs. Briefly, contigs were submitted to the ResFinder webpage while the identity threshold was set to 90% with 60% minimum length match. A custom script (resParse.py) was used to predict the bacterial origin of contigs possessing ARGs and the top candidate was chosen for further analysis.

Resistance to antibacterial biocides and metals were also included as part of the resistome analysis to assess the co-occurrence potentials of BMRGs and ARGs. To

examine this, merged reads were aligned to the BacMet database (version 1.1)[78] using DIAMOND[79] with an E-value cutoff ≤10$^{-10}$. Since resistance genes with transfer potentials are of greater interest than chromosomal genes, analysis of antibacterial biocide and metal resistance protein sequences was restricted to plasmid-borne sequences. A total of 161 metal resistance genes and 37 biocide resistance genes from the BacMet database[78] were detected in the samples and included in this analysis. Data were normalized by 16S rRNA gene as performed with ARGs counts prior to comparisons.

In summary, three independent databases were implemented in this resistome analysis pipeline to assess the profile of all resistance genes, transferrable genes and heavy metal/biocidal resistance genes, respectively. Taxonomy prediction of ARGs was summarized at the family level of bacteria to obtain better accuracy.

**Functional capacity analysis**. To study the functional composition of fecal microbial communities, merged metagenomic sequence reads were annotated using DIAMOND (cutoff E-value ≤10$^{-10}$)[79] against two reference databases, the Carbohydrate-Active EnZymes database (CAZy)[80] and the Clusters of Ortho-logous Groups of proteins database (COG)[81]. Data were normalized using the total number of trimmed reads, and normalized counts were used to calculate the Bray-Curtis dissimilarity between samples. The Bray-Curtis dissimilarity matrix was then used to perform NMDS. In addition, DESeq2[82] was used to estimate fold-change and dispersion over time. Sequencing reads that were predicted to encode CAZy results and identified to be differentially distributed over time (padj <0.01 with Benjamini–Hochberg correction), were aligned to metagenomic-assembled contigs (see above) by using BWA-MEM[83]. Contigs carrying sequences which encode differentially distributed CAZy enzymes were further used to map our custom database including RefSeq[68] and 4941 metagenome-assembled genomes[69] using taxator-tk (version 1.3.3)[77] with parameters -a megan-lca -t 0.3 -e 0.01 to predict the bacterial origin. The most abundant predicted taxon at the family level for each CAZy family was chosen for further analysis.

**Absolute quantification (qPCR) of Enterobacteriaceae and ARGs**. In addition to metagenomic sequencing, quantitative PCR was conducted for two purposes: (1) to determine the absolute abundance of Enterobacteriaceae and (2) to validate the ARG results obtained with sequencing. This independent approach will help to generalize our resistome analyses by assessing more samples ($n = 41$). *E. coli* strain sldp 38-1 was used to generate standard curve for Enterobacteriaceae. Standard curve was prepared using a ten-fold dilution series of DNA isolated from a late exponential phase liquid cultures for which cell numbers were determined by quantitative culture. For validation of ARG results, *tetQ*, one highly abundant tetracycline resistance gene (Supplementary Fig. 6), was chosen to represent the ARGs that gradually increased over time. Ct and delta-Ct values were calculated to assess the dynamic change in the abundance of *tetQ* over time. Primers used for quantitative PCR are listed in Supplementary Table 1. A one-way ANOVA was applied to assess the statistical significance of qPCR results after checking normality.

**Shotgun metagenomic sequencing statistical and network analysis**. Statistical significance for time-series data of ARG/BMRG abundance and richness, and of CAZy family richness was assessed by using a Friedman's test followed by multiple pairwise comparisons using Nemenyi post-hoc test within the PMCMR package (version 4.3) in R. Differences in beta-diversity of fecal resistome and community function (i.e., CAZy and COG analysis) based on a Bray-Curtis dissimilarity cal-culation were tested using adonis2 in the vegan package after checking for dif-ferences in dispersion using betadisper[84].

ARGs predicted from the AMR++ pipeline and BMRGs predicted with the BacMet database were used to model the co-occurrence patterns. Resistance genes with transfer potential that were present in at least 6 (of 12) dairy fecal samples were included in this analysis. A correlation matrix was calculated based on normalized abundance data using Spearman's rank correlation between resistance genes. Only correlations that were strong (rho > 0.8) and significant after Benjamini–Hochberg correction (padj <0.01)[35] were kept in the matrix. The correlation network was visualized using Gephi (0.9.2).

**Ethics approval**. All animal work was approved by the University of California, Davis Animal Care and Use Committee prior to beginning of the experiment under protocol #18540.

**Reporting summary**. Further information on research design is available in the Nature Research Reporting Summary linked to this article.

## Data availability
Sequencing data generated from both amplicon and shotgun metagenomes in this study have been deposited with the NCBI SRA (PRJNA438833) and are publicly available. The source data underlying Figs. 2a, 6b and 10b and Supplementary Fig. 2c are provided as a Source Data file. Other data that support the findings of this study are available from the corresponding author upon request.

## Code availability
A python custom script, resParse.py, based on BLAST, which was used to predict the bacterial origin of metagenomic contigs possessing ARGs estimated from ResFinder database, is freely available at [https://github.com/akre96/ResistBlast].

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

## Acknowledgements

We thank Alice Yu, Karen Kalanetra, Steve Frese, Samir Akre, Nick Jensen, Shannon Snook, Douglas Gisi, and James Moller for helpful discussions and guidance in samples collection, DNA sequencing methodologies and analysis. This work used the Vincent J. Coates

Genomics Sequencing Laboratory at UC Berkeley, supported by NIH S10 Instrumentation Grants S10RR029668 and S10RR027303. This work has been supported by National Institutes of Health awards R01AT007079 and R01AT008759 and the Peter J. Shields Endowed Chair in Dairy Food Science. DGL is funded by the U.S. Department of Agriculture (USDA) project 2032-53000-001-00-D. The USDA is an equal opportunity employer.

## Author contributions

J.L., D.J., E.D., and D.A.M. designed experiments; J.L. and D.J. executed experiments; J.L., D.H.T., M.X.M., M.L.T., and D.G.L. analyzed the data. J.L. and D.A.M. wrote the paper, with input from all authors. All authors read and approved the final manuscript.

## Additional information

**Competing interests:** D.A.M. is a co-founder of Evolve Biosystems, a company focused on diet-based manipulation of the gut microbiota. Evolve Biosystems had no role in the conceptualization, design, data collection, analysis, or preparation of this manuscript. The remaining authors declare no competing interests.

