## [Peer Review File · Nature Communications]

Reviewers' comments:

Reviewer #1 (Remarks to the Author):

The authors present a very well written and coherent metagenomic dataset that describes ARGs in dairy calves in their early stages of life. They identify a "transition" of ARG potential that changes over the lifetime of the experiment, and attempt to connect it to various taxa that are present in the samples. I have the following comments that could be used to strengthen the analysis presented and resulting hypotheses.

Major

- Line 140: "Streptococcaceae, Enterobacteriaceae, and Enterococcaceae, with these combined representing ~90% of the microbiota (Fig. S1c). A substantial portion (~ 30.6%) of the early dairy fecal microbiota (week 1) were comprised of these families (Fig. 1d). These findings clearly suggest that colostrum acts as a carrier of specific bacterial species that seed and temporally colonize the gut microbiome (propagule) of dairy calves." Merely comparing relative abundance patterns at a family-level across two distinct analyses, is perhaps too simplistic to back up the strong statement given in Line 143. Are there shared OTUs /closely related populations that are carried over from the Colostrum to the fecal samples?
- Line 176-179: "suggesting that at least a portion of 179 the diminished ARGs resided in these taxa." To me this seems a strange way to connect the levels of ARGs to specific taxa. Wouldn't phylogenetic analysis of the ARGs to Enterococcaceae-affiliated populations be a more convincing way to do this? Given that the 16S analysis were classified as E.coli, it seems likely to be a taxonomic link.
- Line 291- (CAZy analysis) This comment is in many ways a follow up to my previous one concerning linking function to taxa. The authors state on Line 319: "The changes in the CAZY enzyme abundances suggest that the decrease in ARGs over time is not merely due to the absence of colostrum, but due to the presence of new carbon sources in the diet that drive abundance of taxa that harbor enzymes which can digest plant polysaccharides and are, coincidentally, also low in overall ARGs." It would be much more conclusive if a genome-centric approach using metagenome assembled genomes was employed to connect ARGs, CAZymes etc... to specific populations, and likely taxa
- Line 438-440: What are the main carbohydrates in the oat starter feed? For example, I was surprised not to see CAZymes associated with beta-glucan degradation (i.e. GH16, GH5, GH9). I also think that the enzymes targeting starch appear to decrease at the final timepoints (Figure 9), which somewhat conflicts with authors hypothesis that diet drives changes in CAZyme profiles. More should be done here to connect the CAZyme profile to the specific dietary carbohydrates found in the feed.

Minor

- Line 63: "AMR in livestock limiting therapeutic options and create reservoirs of resistance that can be transmitted to humans via the food chain or environmental effluents." Do you mean "limits"?
- Fig. 2a & b largely show the same thing? Perhaps part b could be moved to sup material?
- Line 158 and Figure 2a: "copies of ARG per 16S rRNA gene each sample" It is not clear how this was determined. This should be included in the Figure legend
- Figure 9: what family is CEO, GTO etc..?

Reviewer #2 (Remarks to the Author):

The authors present a very descriptive study including some 16S analysis and metagenomic analysis of the fecal microbiome of dairy calves. They characterise predicted AMR genes over time

and also look at richness and diversity.

The work would be far more impactful if the researchers could have actually tested the resistance of some of the AMR predictions - many ARGs detected from metagenomics may not actually confer resistance

The authors ignore previous research in the area e.g.

<https://microbiomejournal.biomedcentral.com/articles/10.1186/s40168-017-0378-z>

The authors should discuss the fact that by measuring bacteria in the faeces, they may be measuring dead bacteria!

pg 152: "we expect small resistome variance between subjects" - why?

154: 14.6X sequencing coverage - of what?

178: " suggesting that at least a portion of the diminished ARGs resided in these taxa" this should be possible to test with metagenomic assembly - was this attempted?

237: 15X sequencing coverage of what?

There is a missed opportunity here - assembly and binning could possibly have allowed the authors to track strains from collustrum and into the calves. Why was assembly and binning not attempted?

Did the authors sequence the food/diet ("milk replacer")? this is another source of bacteria

555: what is the purpose of merging reads with FLASH here? What was the library insert size?

557: our tests of MetaPhlAn suggest it is not good with rumen data. Did the authors use any other profiling tool?

558: what does this statement mean? "and the relative abundance of taxa was used to calculate the correlation with ARG abundance over time" ?

Reviewers' comments:

Reviewer #1 (Remarks to the Author):

The authors present a very well written and coherent metagenomic dataset that describes ARGs in dairy calves in their early stages of life. They identify a “transition” of ARG potential that changes over the lifetime of the experiment, and attempt to connect it to various taxa that are present in the samples. I have the following comments that could be used to strengthen the analysis presented and resulting hypotheses.

Response: We thank the reviewer for the positive comments and suggestions.

Major

- Line 140: “Streptococcaceae, Enterobacteriaceae, and Enterococcaceae, with these combined representing ~90% of the microbiota (Fig. S1c). A substantial portion (~30.6%) of the early dairy fecal microbiota (week 1) were comprised of these families (Fig. 1d). These findings clearly suggest that colostrum acts as a carrier of specific bacterial species that seed and temporally colonize the gut microbiome (propagule) of dairy calves.” Merely comparing relative abundance patterns at a family-level across two distinct analyses, is perhaps too simplistic to back up the strong statement given in Line 143. Are there shared OTUs /closely related populations that are carried over from the Colostrum to the fecal samples?

*Response: We appreciate the reviewer’s suggestions. To obtain more convincing evidence of bacterial transmission from colostrum to newborn dairy calves, we relied on analyses of shotgun metagenomic sequenced samples to track bacteria at the strain level. Because Enterobacteriaceae is the microbial clade harboring the most ARGs (Fig. 3c) and *E. coli* represents the majority of sequences classified in this family (Fig. S4), we elected to track the presence of strains belonging to this species in our cohort. We performed the analyses with two independent approaches, 1) PanPhlAn which is a strain-level metagenomic profiling tool that defines strains as unique combinations of genes in the pangenome of a species and can accurately identify the dominant strain of given metagenomes (<https://bitbucket.org/CibioCM/panphlan/wiki/Home>), 2) StrainEst which is another reference-based method but uses the Single Nucleotide Variants (SNV) profiles of the available genomes of selected species (e.g. *E. coli*) to determine the number and identity of coexisting strains in a metagenome (<https://github.com/compmetagen/strainest>).*

The PanPhlAn results indicate that the dominant *E. coli* strain from colostrum and fecal samples at day 2 have very similar genetic profiles (Jaccard; PERMANOVA; $P = 0.15$) (Fig. S2a). At day 5, in the fecal samples of dairy calves, dominant *E. coli* strains start to diverge which eventually result in distinct functional profiles at week 3 (Jaccard; PERMANOVA; $P = 0.004$) (Fig. S2a). Our StrainEst data suggest that there was a higher diversity of *E. coli* strains in dairy calf at day 2 compared to colostrum samples, and this diversity increased by day 5 but decreased dramatically at week 3 (Fig. S2b). Specifically, 63 distinct *E. coli* strains on average were observed in colostrum, and

33.13% of them are shared with dairy calves at day 2, 15.63% of them are shared with dairy calves at both day 2 and day 5, and only 3.08% of *E. coli* strains in colostrum were kept until week 3 in one (out of three) fecal sample (Fig. S2c). For both strain profiling analyses, *E. coli* was detected in a subset of sequenced samples at week 3 and was below our detection limit at metagenomes from week 7. This is likely due to the gradually increased gut microbiome diversity and decreased relative abundance of *E. coli* over time. Overall, our findings are consistent with the original hypothesis that colostrum acts, in part, as a carrier of specific bacterial species that seed and temporally colonize the gut microbiome (propagule) of dairy calves.

Finally, we argue that some bacterial species are very recombinogenic (e.g. *E. coli*) (Welch et al., *Extensive mosaic structure revealed by the complete genome sequence of uropathogenic Escherichia coli*. PNAS. 2002) and transmission of antibiotic resistance can occur at the gene level, for example, bacteria can exchange genes through horizontal gene transfer. In light of these possibilities occurring in the natural ecosystem, tracking bacterial strains provides basis but can only partially explain the antibiotic resistance dissemination while results at the ARG level is more conclusive (Fig. 6).

Revisions made: We have followed the reviewer's suggestion and metagenomic strain profiling of *E. coli* was performed in colostrum and fecal samples. Supplemental figure 2 was added in the revised manuscript and the results were updated in line 149-179.

- Line 176-179: "suggesting that at least a portion of 179 the diminished ARGs resided in these taxa." To me this seems a strange way to connect the levels of ARGs to specific taxa. Wouldn't phylogenetic analysis of the ARGs to Enterococcaceae-affiliated populations be a more convincing way to do this? Given that the 16S analysis were classified as *E. coli*, it seems likely to be a taxonomic link.

Response: We agree with the reviewer that the correlation between the relative abundance of specific taxa and normalized ARG abundance is not the ideal way to connect bacteria and AR genes. Instead of working with short (raw) reads, in the current revision, we applied MEGAHIT to assemble our sequencing reads into metagenomic contigs with default settings. Next, the ARG-aligned sequencing reads (see AMR++ pipeline in method) were used to align to these contigs with BWA-MEM, and contigs possessing ARG sequences were filtered and kept for taxonomic assignment. taxator-tk (<https://github.com/fungs/taxator-tk>), a software which is designed to perform taxonomic analysis of assembled metagenomes was applied to predict the bacterial origin of specific contigs. Specifically, we used taxator-tk to against RefSeq with parameters -a megan-lca -t 0.3 -e 0.01 to assign taxonomy of ARG-containing contigs at the family level. The results are consistent with our original hypothesis and became more conclusive. Specifically, ARGs observed in fecal samples of dairy calves were predicted to originate from 81 different bacterial families with Enterobacteriaceae harboring the most ARGs for both abundance and richness. Thank you for your insightful comments.

Revisions made: The updated taxonomy of ARGs observed in dairy fecal samples were added in the revised manuscript as Fig. 3c (ARG normalized abundance) and Fig. 3d (ARG richness/diversity distribution over time). Text was updated in line 215-228.

- Line 291- (CAZy analysis) This comment is in many ways a follow up to my previous one concerning linking function to taxa. The authors state on Line 319: “The changes in the CAZy enzyme abundances suggest that the decrease in ARGs over time is not merely due to the absence of colostrum, but due to the presence of new carbon sources in the diet that drive abundance of taxa that harbor enzymes which can digest plant polysaccharides and are, coincidentally, also low in overall ARGs.” It would be much more conclusive if a genome-centric approach using metagenome assembled genomes was employed to connect ARGs, CAZymes etc... to specific populations, and likely taxa

Response: We thank the reviewer again for the insightful suggestion. In the original submission, we initially predicted the bacterial origin of CAZy enzymes by analyzing short sequencing reads. In this revision, we applied the same approach to connect both CAZy enzymes and ARGs to bacterial taxa, that is, we assigned taxonomy to assembled contigs which encode differentially abundant CAZy enzymes (see Methods for details). In brief, CAZy-aligned reads were aligned to metagenome assembled contigs, and contigs were filtered and the ones containing CAZy sequences were kept to run through taxator-tk. The taxonomy was characterized at the family level for better accuracy. Our new analyses is consistent with findings in our original submission. In particular, Enterobacteriaceae was predicted to contribute most of the CAZy enzymes that exhibited higher abundance during early days and decreased at later time points (CE8-GH108 and GH127-GT51) (Fig. 10b). In contrast, CAZy enzymes, GH20-GH30 and CE6-GH27, which showed higher abundance in later time points were predicted to mostly originate from Bacteroidaceae (Fig. 10b).

Revisions made: We followed the reviewer’s suggestion and Fig. 10 (Fig. 9 in the initial submission) has been updated with the new analyses.

- Line 438-440: What are the main carbohydrates in the oat starter feed? For example, I was surprised not to see CAZymes associated with beta-glucan degradation (i.e. GH16, GH5, GH9). I also think that the enzymes targeting starch appear to decrease at the final timepoints (Figure 9), which somewhat conflicts with authors hypothesis that diet drives changes in CAZyme profiles. More should be done here to connect the CAZyme profile to the specific dietary carbohydrates found in the feed.

Response: The major ingredients in the calf starter of our study are corn, soybean, barley, wheat, oat and rice (see Methods). The main carbohydrates are starch which represents roughly 57-77% of ingredients in the calf starter. (*Khan et al., Starch Source Evaluation in Calf Starter: I. Feed Consumption, Body Weight Gain, Structural Growth, and Blood Metabolites in Holstein Calves, Journal of Dairy Science, 2007.*). We named/mentioned the calf starter several time as “oat starter” in the original submission,

which is inaccurate and confusing. The correct name is calf starter and the revised manuscript has been updated accordingly.

We thank the reviewer for the great point on beta-glucan in calf starter. The calf starter, which were used in this study, was supplied from a local company “Associated Feed & Supply Co., Turlock, CA” and it has a guaranteed analysis of crude fiber less than 4%. We also learned that of the common fiber (wheat, rye, oats and barley), the largest amounts of β -glucan are typically found in barley (3-11%) and oats (3-7%) (http://www5.agr.gc.ca/resources/prod/doc/misb/fb-ba/nutra/pdf/B-Glucans_Eng.pdf; Wood, P.J. and M.U. Beer. 1998. In: *Functional Foods. Biochemical & Processing Aspects.* Mazza, G. (Ed). Technomic Publication Company. Inc. Lancaster, PA pp. 1-37.). That is, the percentage of beta-glucan in calf starter should be between 0.12- 0.44% at most. Even though we believe there is a gradual increase of calf starter intake over time, given that the percentage of beta-glucan in the starter is so low it is likely that the change of associated CAZy enzymes do not reach statistical significance.

We apologize for the confusion, but the abundance of observed CAZy enzyme targeting starch actually increased gradually over time. Specifically, two GH families, family GH97, which includes glucoamylase (EC 3.2.1.3) and α -glucosidase (EC 3.2.1.20)), and family GH57, which includes α -amylase (EC 3.2.1.1), increased significantly over time (Fig. 10 in the revised manuscript).

Revisions made: We thank the reviewer’s insightful comments on this analysis. We modified “oat starter” to “calf starter” throughout the manuscript. Text has been updated in line 530-532.

Minor

- Line 63: “AMR in livestock limiting therapeutic options and create reservoirs of resistance that can be transmitted to humans via the food chain or environmental effluents.” Do you mean “limits”?

Response: We thank the reviewer for catching this error.

Revisions made: The text has been updated in line 63.

- Fig. 2a & b largely show the same thing? Perhaps part b could be moved to supplemental material?

Response: We concur and thank for the reviewer’s comment.

Revisions made: We elected to remove the original Fig. 2b, and Fig 2 has been updated accordingly.

- Line 158 and Figure 2a: “copies of ARG per 16S rRNA gene each sample” It is not clear how this was determined. This should be included in the Figure legend

Response: We apologize for the confusion.

Revisions made: We added a note to the Fig. 2 legend as “The counts data were normalized by 16S rRNA gene and the ARG abundance was expressed as “copy of ARG per copy of 16S rRNA gene” (see Methods).”.

- Figure 9: what family is CE0, GT0 etc..?

Response: We apologize for the confusion. Families CBM0, CE0 and GT0 refer to unclassified CBMs, CEs and GTs respectively in the CAZy database download 07152016 (<http://csbl.bmb.uga.edu/dbCAN/index.php>).

Revisions made: We added a note to the Fig. 10 legend to clarify this.

Reviewer #2 (Remarks to the Author):

The authors present a very descriptive study including some 16S analysis and metagenomic analysis of the fecal microbiome of dairy calves. They characterise predicted AMR genes over time and also look at richness and diversity.

The work would be far more impactful if the researchers could have actually tested the resistance of some of the AMR predictions - many ARGs detected from metagenomics may not actually confer resistance

Response: We agree with the reviewer’s comment. Genotypes do not necessarily guarantee phenotypes. Ideally, we’d like to isolate bacteria from feces and test the antibiotic resistance phenotype accordingly. However, low quantity of fecal samples during early life (e.g. day 2) on a swab limited our ability to do so after extracting DNA.

Revisions made: We followed the reviewer’s suggestion and discussed this limitation in line 507 to 511.

The authors ignore previous research in the area e.g.
<https://microbiomejournal.biomedcentral.com/articles/10.1186/s40168-017-0378-z>

Response: We sincerely apologize for this omission in our original submission. Actually, this prior study nicely identified correlations between diet and resistome in bovine which provides the rationale of our work.

Revisions made: We appreciate the reviewer’s comment, and the study published by Auffret et al. was cited in our introduction (line 86-92).

The authors should discuss the fact that by measuring bacteria in the faeces, they may be measuring dead bacteria!

Response: Thank you and we agree with the reviewer. We view this as the natural limitations of an analyses which relies on high-throughput sequencing and have provided a comment on this issue. We also note that, naturally competent bacteria are able to uptake naked DNA particles including those released from dead microbes (*Blokesch et al., Natural competence for transformation. Curr Biol 26, 3255. 2016.*). While we do not know the probability of this event occurring, uptake of DNA from dead bacteria has the potential to contribute to ARG transmission. We also addressed this point in the discussion.

Revisions made: We followed the reviewer's suggestion and discussed this in line 507 to 514.

pg 152: "we expect small resistome variance between subjects" - why?

Response: Our argument is based on the fact, in our cohort, dairy calves of the same breed are housed in the same way with identical diet, and at the same age they have very similar microbiota. It is widely accepted that microbial taxonomy greatly shapes the resistome structure (*Forsberg et al., Bacterial phylogeny structures soil resistomes across habitats. Nature. 2014; Pärnänen et al., Maternal gut and breast milk microbiota affect infant gut antibiotic resistome and mobile genetic elements. Nature Communications. 2018.*). We initially selected samples for metagenomic sequencing based this assumption, and as a result, we observed small resistome variances between calves and our sequencing sample size did offer us sufficient statistical power to capture the dynamic changes of ARGs abundance over time (effect size is about 6.68-fold decrease of ARG abundance from day 2 to week 7). (Fig. 3a).

Revisions made: We updated the text in line 184-188 as "Sequencing sample size (n=3 dairy calves) was determined based on the fact that dairy calves of the same breed in our cohort have limited genetic diversity, and they were housed in the same way with identical diets, more importantly, at the same age they indicate very similar microbiota (16S rRNA marker gene sequencing analysis) and thus we expect small resistome variance between subjects", and in line 590-594 as "Dairy calves at the same age indicated limited variances of the gut microbiota, and bacterial phylogeny greatly shapes the microbial antibiotic resistome, and thus we expect a comparable fecal resistome between subjects. Consequently, a number of three calves would allow us to observe the dynamic changes of ARGs abundance over time if any."

154: 14.6X sequencing coverage - of what?

Response: We apologize for the confusion, but this is the sequencing coverage we estimated of assembled metagenomic contigs. We assembled the metagenomes using MEGAHIT and raw sequencing reads were used to map to assembled contigs and sequencing coverage was estimated using BMap (<https://jgi.doe.gov/data-and-tools/bbtools/bb-tools-user-guide/bbmap-guide/>).

Revisions made: The text has been updated in line 189-190 to clarify this.

178: " suggesting that at least a portion of the diminished ARGs resided in these taxa" this should be possible to test with metagenomic assembly - was this attempted?

Response: We agree with the reviewer that this can be definitely tested with metagenomic assembly. In this revision, we followed this approach to assign taxonomy to ARGs. Instead of working with short (raw) reads, in the current revision, we applied MEGAHIT to assemble our sequencing reads into metagenomic contigs with default settings. Next, the ARG-aligned sequencing reads (see AMR++ pipeline in method) were used to align to these contigs with BWA-MEM, and contigs possessing ARG sequences were filtered and kept for taxonomic assignment. taxator-tk (<https://github.com/fungs/taxator-tk>), a software which is designed to perform taxonomic analysis of assembled metagenomes was applied to predict the bacterial origin of specific contigs. Specifically, we used taxator-tk to against RefSeq with parameters -a megan-lca -t 0.3 -e 0.01 to assign taxonomy of ARG-containing contigs at the family level. The results are consistent with our original hypothesis and became more conclusive. Specifically, ARGs observed in fecal samples of dairy calves were predicted to originate from 81 different bacterial families with Enterobacteriaceae harboring the most ARGs for both abundance and richness. Thank you for your insightful comments.

Revisions made: The updated taxonomy of ARGs observed in dairy fecal samples were added in the revised manuscript as Fig. 3c (ARG normalized abundance) and Fig. 3d (ARG richness/diversity distribution over time). Text was updated in line 215-228.

237: 15X sequencing coverage of what?

Response: Please see previous response. We appreciate the reviewer's comment.

Revisions made: The text has been updated in line 280-281 to clarify this.

There is a missed opportunity here - assembly and binning could possibly have allowed the authors to track strains from colostrum and into the calves. Why was assembly and binning not attempted?

Response: We agree with the reviewer that tracking bacterial strains from colostrum to calves can definitely strengthen our analyses. As metagenomic assembly and binning is typically sensitive to sequencing depth/coverage, and some taxa (e.g. Enterobacteriaceae) which are predict to harbor lots ARGs had very low relative abundance after week 1, we elected to address these concerns with a short-read-alignment-based approach. Specifically, in this revision, analyses were performed by using two independent pipelines, 1) PanPhlAn which is a strain-level metagenomic profiling tool defines strains as unique combinations of genes in the pangenome of a species and can accurately identify the dominant strain of given metagenomes (<https://bitbucket.org/CibioCM/panphlan/wiki/Home>), 2) StrainEst which is another

reference-based method but uses the Single Nucleotide Variants (SNV) profiles of the available genomes of selected species (e.g. *E. coli*) to determine the number and identity of coexisting strains in a metagenome (<https://github.com/compmetagen/strainest>). Because Enterobacteriaceae is the microbial clade harboring the most ARGs (Fig. 3b) and *E. coli* represents the majority of sequences classified in this family (Fig. S4), we elected to track the presence of this species in our cohort.

The PanPhlAn results indicate that the dominant *E. coli* strain from colostrum and fecal samples at day 2 have very similar genetic profiles (Jaccard; PERMANOVA; $P = 0.15$) (Fig. S2a). At day 5, in the fecal samples of dairy calves, dominant *E. coli* strains start to diverge which eventually demonstrate distinct functional profiles at week 3 (Jaccard; PERMANOVA; $P = 0.004$) (Fig. S2a). Our StrainEst data suggest that there was a higher diversity of *E. coli* strains in dairy calf at day 2 compared to colostrum samples, and this diversity increased by day 5 but decreased dramatically at week 3 (Fig. S2b). Specifically, 63 distinct *E. coli* strains on average were observed in colostrum, and 33.13% of them are shared with dairy calves at day 2, 15.63% of them are shared with dairy calves at both day 2 and day 5, and only 3.08% of *E. coli* strains in colostrum were kept until week 3 in one (out of three) fecal sample (Fig. S2c). For both strain profiling analyses, *E. coli* was detected in a subset of sequenced samples at week 3 and was below our detection limit at metagenomes from week 7. This is likely due to the gradually increased gut microbiome diversity and decreased relative abundance of *E. coli* over time. Overall, our findings are consistent with the original hypothesis that colostrum acts as a carrier of specific bacterial species that seed and temporally colonize the gut microbiome (propagule) of dairy calves.

Finally, we argue that some bacterial species are very recombinogenic (e.g. *E. coli*) (Welch et al., *Extensive mosaic structure revealed by the complete genome sequence of uropathogenic Escherichia coli*. PNAS. 2002) and transmission of antibiotic resistance can occur at the gene level, for example, bacteria can exchange genes through horizontal gene transfer. In light of these possibilities happening in the natural ecosystem, tracking bacterial strains provides basis but can only partially explain the antibiotic resistance dissemination while results at the ARG level is more conclusive (Fig. 6). We sincerely thank you for the opportunity to address this concern, and manuscript has been updated accordingly.

Revisions made: We have followed the reviewer's suggestion and metagenomic strain profiling of *E. coli* was performed in colostrum and fecal samples. Supplemental figure 2 was added in the revised manuscript and the results were updated in line 149-179.

Did the authors sequence the food/diet ("milk replacer")? this is another source of bacteria

Response: We thank the reviewer for the insightful comment. Yes, milk replacer and other environmental settings are definitely potential sources of bacteria which can be

transmitted to dairy calves. Our strain profiling analyses with *E. coli* suggested there were approximately 78 distinct strains observed in dairy calves at day 2, but the colostrum-feces shared *E. coli* strains only represent 33.13% of the ones in colostrum. This suggests that there must be bacteria from other sources contributing to the initial assembly of dairy gut microbiome. However, our original experiment focuses on studying the contribution of colostrum, the first food to dairy calves, in seeding the gut microbiome and resistome, and unfortunately, samples from other sources were not collected during the period when the study was conducted. We argue that antibiotic resistance transmission occurs via multiple routes, and both bacterial transmission/expansion and horizontal gene transfer could contribute to this effect. We identified a very similar ARG composition in both colostrum and dairy calves at day 2, and over 90% of ARGs from calves are shared with colostrum (Fig. 6), and thus we argue that our conclusion about “Colostrum seeds the vast majority of early-life ARGs in dairy calves” is valid.

Revisions made: We appreciate the reviewer’s comments and acknowledged the role of other food sources as well as environmental factors in transmitting bacteria and ARGs in the discussion (line 445 to 448).

555: what is the purpose of merging reads with FLASH here? What was the library insert size?

Response: Our library insert sizes averaged at 200-225bp, and the sequencing was completed using the Illumina HiSeq 4000 with 150 paired-end reads. The median size of our merged metagenomic sequencing reads is 190bp. FLASH (<https://ccb.jhu.edu/software/FLASH/>) is designed to merge paired sequencing reads and the resulting longer reads can significantly improve metagenome assemblies (*Magoc et al., FLASH: fast length adjustment of short reads to improve genome assemblies. 2011. Bioinformatics.*). Also, merging allowed us to assign ARGs to longer reads with better accuracy. In addition, mapping short reads to protein databases (e.g. CAZy), unlike genomes, does not leverage pairs of reads so it is more effective to merge overlapping reads into a single longer read before mapping to a protein database.

557: our tests of MetaPhlAn suggest it is not good with rumen data. Did the authors use any other profiling tool?

Response: We apologize for the confusion, but we actually applied MetaPhlAn2 instead of MetaPhlAn in our study. Also, the samples we analyzed are fecal samples collected from dairy calf rectum which differ from rumen content samples (*Dill-McFarland et al., Microbial succession in the gastrointestinal tract of dairy cows from 2 weeks to first lactation. Scientific Reports. 2017.*). In our initial submission, we used MetaPhlAn2 to profile the taxonomic composition of metagenomes at both family and species levels. Next, the relative abundance of bacterial families was used to correlated with the dynamic changed ARG abundance to further predict/correlate the bacterial origin of ARGs. In the current revision, we accomplished this inference (bacterial origin of ARGs) in a more convincing approach (as the reviewer suggested), specifically, the bacterial

taxa were assigned to metagenomic assembled contigs which possess ARGs (see above). Thus, we elected to remove the original correlation analyses (Fig. 3b in the original submission).

The metagenomes were also profiled at the species level to specifically assess the relative abundance of *E. coli* (Fig. S4 in the current revision). To validate this result, we applied an independent program kraken2 (<https://ccb.jhu.edu/software/kraken2/>) which relies on the RefSeq database to assign taxonomy to metagenome sequencing reads. The relative abundance of *E. coli* was further estimated using bracken (<https://ccb.jhu.edu/software/bracken/>). The results from two pipelines, MetaPhlAn2 and kraken2 were compared as follow. In particular, the relative abundance of *E. coli* from metagenomes predicted from kraken2 was slightly higher at the beginning than that from metaphlan2, but they were very similar and there was no overall significant difference ($P = 0.45$) (left). The *E. coli* relative abundance estimated from two pipelines are significantly correlated ($\rho = 0.97$, $P < 0.001$) (right). In light of these results, we kept results from MetaPhlAn2 presented in Fig. S4.

We thank the reviewer for the comment, and we would very much like to hear your specific concerns/limitations on MetaPhlAn2 for profiling rumen metagenomes.

558: what does this statement mean? "and the relative abundance of taxa was used to calculate the correlation with ARG abundance over time" ?

Response: In our initial submission, we calculated the correlation matrix between the relative abundance of bacterial taxa at the family level and the normalized ARG abundance over time. Significant correlations were shown in our original Fig. 3b, and we inferred such microbial clades are likely the taxa which harbor the most ARGs. While with the new analyses performed (see above), the original results were unnecessary and removed in the revised manuscript. We apologize for the confusion.

Reviewers' comments:

Reviewer #1 (Remarks to the Author):

[No further comments for authors.]

Reviewer #2 (Remarks to the Author):

> On sequence coverage (14.6X coverage of the MEGAHIT contigs)

This makes no sense as a single figure, I think. In a metagenomic context, each contig will have different coverage based on the abundance of the host genome in the sample. I just don't think quoting an average coverage across the entire assembly really tells us anything.

> on MEGAHIT assembly

Whilst I welcome the assembly of reads into larger contigs, and the subsequent taxonomic assignment of those contigs, I still think there is a missed opportunity of trying to bin these contigs into genomes.

Whether binned or not, it would be interesting to compare the assemblies to:

<https://www.nature.com/articles/s41467-018-03317-6>

<https://www.nature.com/articles/s41564-017-0012-7>

<https://www.ncbi.nlm.nih.gov/pubmed/28731473>

<https://www.nature.com/articles/s41564-018-0225-4>

<https://www.biorxiv.org/content/10.1101/489443v1>

By comparing the assemblies only to RefSeq, the authors will miss out on many of the above metagenome assembled genomes which only appear in the GenBank genomes section of the NCBI database. See file:

ftp://ftp.ncbi.nlm.nih.gov/genomes/genbank/bacteria/assembly_summary.txt

ftp://ftp.ncbi.nlm.nih.gov/genomes/genbank/archaea/assembly_summary.txt

The strain profiling of E coli is a welcome addition, but it is still not clear to me why binning of the assemblies was not attempted, either with each sample individually, or using a co-assembly (to increase coverage)

> on colostrum and diet (milk replacer)

It is an important point about the environment. If ARG transmission is only from mother to calf via colostrum then that is quite important; however if ARG transmission happens via the environment or the feed/milk, then that is also important as it will lead to ARG transmission between herds.

I am not convinced that without sequencing the environment and diet (milk replacer) that we can be sure there is no environmental transmission

> on the use of FLASH

I can certainly see the benefit of using FLASH before mapping of reads, but using it before assembly with MEGAHIT (a de bruijn assembler) will likely have no positive benefits and many

negative benefits.

> MetaPhlan2 and Kraken2 results

It is of no surprise that these two agree as they are based on the same database - i.e. RefSeq. However this database is fundamentally lacking many cattle-derived bacteria. We have many results showing that these two tools are not accurate for cattle derived data due to the lack of good references in the databases.

See figure 4 in <https://www.nature.com/articles/s41467-018-03317-6>

What the authors can do is take a set of cattle-derived bacteria, simulate reads from them, and run Kraken2/MetaPhlan2 to see whether those two tools can recapitulate the input data. I think the authors will find they are quite inaccurate.

Reviewers' comments:

Reviewer #1 (Remarks to the Author):

[No further comments for authors.]

Response: We thank the reviewer for the support on publishing this study.

Reviewer #2 (Remarks to the Author):

> On sequence coverage (14.6X coverage of the MEGAHIT contigs)

This makes no sense as a single figure, I think. In a metagenomic context, each contig will have different coverage based on the abundance of the host genome in the sample. I just don't think quoting an average coverage across the entire assembly really tells us anything.

Response: We thank the reviewer for this great point. Our initial attempt was to describe the overall sequencing, but we agree that a single value on average coverage was not very helpful. In this revision, we followed published studies (Stewart et al., Assembly of 913 microbial genomes from metagenomic sequencing of the cow rumen, Nature Communications, 2018; Stewart et al., The genomic and proteomic landscape of the rumen microbiome revealed by comprehensive genome-resolved metagenomics, bioRxiv preprint, 2018) to include the number of sequencing reads and file size of sequencing data instead.

Revisions made: The texts have been updated in line 188-190 and 278-280. We appreciate the reviewer's comment and the opportunity to make our sequencing description clear.

> on MEGAHIT assembly

Whilst I welcome the assembly of reads into larger contigs, and the subsequent taxonomic assignment of those contigs, I still think there is a missed opportunity of trying to bin these contigs into genomes.

Whether binned or not, it would be interesting to compare the assemblies to:

<https://www.nature.com/articles/s41467-018-03317-6>

<https://www.nature.com/articles/s41564-017-0012-7>

<https://www.ncbi.nlm.nih.gov/pubmed/28731473>

<https://www.nature.com/articles/s41564-018-0225-4>

<https://www.biorxiv.org/content/10.1101/489443v1>

By comparing the assemblies only to RefSeq, the authors will miss out on many of the above metagenome assembled genomes which only appear in the GenBank genomes

section of the NCBI database. See file:

ftp://ftp.ncbi.nlm.nih.gov/genomes/genbank/bacteria/assembly_summary.txt

ftp://ftp.ncbi.nlm.nih.gov/genomes/genbank/archaea/assembly_summary.txt

The strain profiling of E coli is a welcome addition, but it is still not clear to me why binning of the assemblies was not attempted, either with each sample individually, or using a co-assembly (to increase coverage)

Response:

1. We thank the reviewer for pointing us to the resources of recently assembled metagenome-resolved genomes. Stewart *et al.* reported that inclusion of 913 RUGs (Rumen Uncultured Genomes) with the RefSeq database improves metagenomic read classification by sevenfold for rumen samples (Stewart *et al.*, Assembly of 913 microbial genomes from metagenomic sequencing of the cow rumen, Nature Communications, 2018). Most recently, the same research group presented the assembly of 4941 RUGs which is currently the largest dataset for rumen-associated uncultured genomes (Stewart *et al.*, The genomic and proteomic landscape of the rumen microbiome revealed by comprehensive genome-resolved metagenomics, bioRxiv).

In this revision, as the reviewer suggested, we included the genome sequences of 4941 RUGs (Stewart *et al.*, 2018, bioRxiv) together with the most recently released RefSeq (v93) into our taxonomy assignment for ARG-containing contigs using taxator-tk (see Methods). The addition of these new sequences/genomes indeed significantly improved our classification rate, and in particular, up to 27% more contigs were assigned a bacterial taxon at the family level (see details in the following figure). The results of modified taxonomy classification have been updated accordingly, and this new analysis further strengthened our argument. For example, in Fig. 3c, we observed a gradually decreased abundance of ARGs from Enterobacteriaceae, but the predicted ARGs in Ruminococcaceae, Bacteroidaceae and Others increased after day2; these findings are consistent with a dynamic gut resistome during early life of dairy calves and the bacterial taxonomy played a significant role in shaping the structure. We sincerely thank the reviewer for the opportunity to improve our taxonomy assignment which greatly improved the quality of this work.

2. Metagenomic binning was applied to single-sample assemblies following a published pipeline (Stewart et al., Assembly of 913 microbial genomes from metagenomic sequencing of the cow rumen, Nature Communications, 2018), and we agree with the reviewer that comparing nearly-complete metagenomic assembled genomes across samples would help to assess bacterial transmission. When binning, we were able to obtain a total of 339 bins from all samples in this cohort, including colostrum and fecal samples. Unfortunately, only 63 of these bins passed the quality threshold of >80% completeness and <10% contamination. Furthermore, none of bins from day 2 fecal samples passed this quality step. As a result, we were unable to compare the bins from colostrum with the samples obtained closest in time to colostrum feeding, or to determine if the bins from the early life samples matched those downstream. There was a high percentage of host-associated DNA are present in early feces which likely reduced the sequencing coverage of microbe DNA and resulted in lower quality bins. Co-assembly would help to increase coverage, but that increases the risk that a single bin would contain reads from closely related species at different time points creating an artificial appearance of transmission.

The problem of closely related organisms binning together may also apply to assemblies performed only on a single sample. As a result, we argue that metagenomic assembly and binning is probably not the best approach to assess the possibility of strain transmission. This is mostly because our inability to distinguish closely-related microbes (i.e. strains) in complex microbial samples. Please refer this argument to following quotes from published literatures:

“The bins are sometimes referred to as ‘population genomes’, as the unsupervised binning usually cannot distinguish the genetic content of closely related organisms (strains) in complex microbial communities.” (Breitwieser et al., A review of methods and databases for metagenomic classification and assembly. Briefings in Bioinformatics. 2017)

“Related species or subspecies introduce extensive overlaps in a kmer set, and therefore create assembly graphs that are considerably more complex as multiple genomes occupy much of the same kmer space. Branches or loops between these homologous regions make traversing the graph more complex, and if either species occurs with a low abundance, then identifying the presence of separate species will be difficult and deconvolving the graph is extremely complex. Mistakes at this point can lead to chimeric contigs containing sequence from more than one (sub-)species and a failure to capture the true diversity of the sample.” (*Ayling et al., New approaches for metagenome assembly with short reads. Briefings in Bioinformatics. 2019*)

Because of our concerns about the low quality of our day 2 bins and the tendency of binning to group closely related species and subspecies, we elected to continue to assess the possibility of strain transmission using PanPhlAn and StrainEst (see Methods). As the reviewer stated, we do believe that we have sufficient data to infer that transmission actually occurred between colostrum and dairy calves. We also believe the addition of a binning-based analysis will be a weaker approach to studying transmission than that already present in the paper.

Revisions made: We have followed the reviewer’s suggestion and used the additional recommended RUGs for the taxonomic analysis. Methods have been updated in line 608-612, line 667-670, and line 709-712. Results related to bacterial taxonomy assignment of contigs have been updated in Fig. 3c&d and Fig. 10b.

> on colostrum and diet (milk replacer)

It is an important point about the environment. If ARG transmission is only from mother to calf via colostrum then that is quite important; however if ARG transmission happens via the environment or the feed/milk, then that is also important as it will lead to ARG transmission between herds.

I am not convinced that without sequencing the environment and diet (milk replacer) that we can be sure there is no environmental transmission

Response: We agree with the reviewer that the housing environment is a critical source of ARGs, and work assessing the possibility of ARG transmission between environment and dairy calves is very important. Unfortunately, the current study focuses on ARG changes in early life with an assessment of colostrum, the first food to dairy calves, in seeding the gut microbiome and resistome. Samples from other sources (water, weaning diet, built-environment, soil, air etc.) were not collected. While related to this topic, we feel a full examination of the environmental factors potentially involved in ARG transfer here is beyond the scope of this focused study.

Revisions made: We appreciate the reviewer’s comments and these limitations have been discussed in line 444 and 447.

> on the use of FLASH

I can certainly see the benefit of using FLASH before mapping of reads, but using it before assembly with MEGAHIT (a de bruijn assembler) will likely have no positive benefits and many negative benefits.

Response: We apologize for the confusion, but the reads used for assembly are **un-merged**. We agree with the reviewer that merged reads are not appropriate for metagenomic assembly.

Revisions made: The text has been updated in line 613-614 to clarify this. Thank you for letting us know we were unclear on this point.

> MetaPhlan2 and Kraken2 results

It is of no surprise that these two agree as they are based on the same database - i.e. RefSeq. However this database is fundamentally lacking many cattle-derived bacteria. We have many results showing that these two tools are not accurate for cattle derived data due to the lack of good references in the databases.

See figure 4 in <https://www.nature.com/articles/s41467-018-03317-6>

What the authors can do is take a set of cattle-derived bacteria, simulate reads from them, and run Kraken2/MetaPhlan2 to see whether those two tools can recapitulate the input data. I think the authors will find they are quite inaccurate.

Response: We agree with the reviewer that the default databases (e.g. RefSeq) use for Kraken2 and MetaPhlan2 are not designed/optimized for rumen data taxonomy profiling. In this revision, the recently assembled 4941 RUGs (see above) were added into the database for a re-run with Kraken2. The relative abundance of *E. coli* predicted from the new analysis (with RUGs in the database) is slightly lower than that predicted from RefSeq itself (see figure below). These results make perfect sense because the new addition of 4941 RUGs are dominated by Ruminococaceae (1111), Lachnospiraceae (640) and Prevotellaceae (521), and the relative abundance of *E. coli* was squeezed when more other reads are assigned to a different taxon. Again, we appreciate the reviewer's guidance to include these great resources of rumen genomes into our work.

Revisions made: We have followed the reviewer's suggestion to include more rumen associated genomes in our analysis, and the updated Enterobacteriaceae species taxonomy composition has been shown in Fig. S4.